# Lineage frequency time series reveal elevated levels of genetic drift in SARS-CoV-2 transmission in England

QinQin Yu[1¤*], Joao A. Ascensao[2‡], Takashi Okada[1,3,4,5‡], The COVID-19 Genomics UK (COG-UK) Consortium[¶], Olivia Boyd[6], Erik Volz[6¶], Oskar Hallatschek[1,3,7*]

1 Department of Physics, University of California, Berkeley, California, United States of America,
2 Department of Bioengineering, University of California, Berkeley, California, United States of America,
3 Department of Integrative Biology, University of California, Berkeley, California, United States of America,
4 Institute for Life and Medical Sciences, Kyoto University, Kyoto, Japan, 5 RIKEN iTHEMS, Wako, Saitama, Japan, 6 MRC Centre for Global Infectious Disease Analysis, Department of Infectious Disease Epidemiology, Imperial College London, London, United Kingdom, 7 Peter Debye Institute for Soft Matter Physics, Leipzig University, Leipzig, Germany

¤Current address: Department of Immunology and Infectious Diseases, Harvard T. H. Chan School of Public Health, Boston, Massachusetts, United States of America
‡ These authors contributed equally to this work.
¶ Membership of The COVID-19 Genomics UK (COG-UK) Consortium is provided in S1 Acknowledgments.
* qinqinyu@berkeley.edu (QY); ohallats@berkeley.edu (OH)

**Data Availability Statement:** Data and code to reproduce the analyses in this manuscript are available at https://github.com/qinqin-yu/sars-cov-2_genetic_drift Code for creating the publicly

## Abstract

Genetic drift in infectious disease transmission results from randomness of transmission and host recovery or death. The strength of genetic drift for SARS-CoV-2 transmission is expected to be high due to high levels of superspreading, and this is expected to substantially impact disease epidemiology and evolution. However, we don't yet have an understanding of how genetic drift changes over time or across locations. Furthermore, noise that results from data collection can potentially confound estimates of genetic drift. To address this challenge, we develop and validate a method to jointly infer genetic drift and measurement noise from time-series lineage frequency data. Our method is highly scalable to increasingly large genomic datasets, which overcomes a limitation in commonly used phylogenetic methods. We apply this method to over 490,000 SARS-CoV-2 genomic sequences from England collected between March 2020 and December 2021 by the COVID-19 Genomics UK (COG-UK) consortium and separately infer the strength of genetic drift for pre-B.1.177, B.1.177, Alpha, and Delta. We find that even after correcting for measurement noise, the strength of genetic drift is consistently, throughout time, higher than that expected from the observed number of COVID-19 positive individuals in England by 1 to 3 orders of magnitude, which cannot be explained by literature values of superspreading. Our estimates of genetic drift suggest low and time-varying establishment probabilities for new mutations, inform the parametrization of SARS-CoV-2 evolutionary models, and motivate future studies of the potential mechanisms for increased stochasticity in this system.

available COG-UK phylogenetic trees is available at https://github.com/virus-evolution/phylopipe.

**Funding:** This material is based upon work supported by the National Science Foundation (NSF, https://www.nsf.gov/) Graduate Research Fellowship under Grant No. DGE 1106400 (to QY), and Japan Society for the Promotion of Science (https://www.jsps.go.jp/english/) KAKENHI (#22K03453, #JP22K06347, to TO). JAA acknowledges support from an NSF Graduate Research Fellowship and a Berkeley Fellowship (https://www.berkeley.edu/). OH acknowledges support by a Humboldt Professorship of the Alexander von Humboldt Foundation (https://www.humboldt-foundation.de/en/). COG-UK is supported by funding from the Medical Research Council (MRC, https://www.ukri.org/councils/mrc/) part of UK Research & Innovation (UKRI), the National Institute of Health Research (NIHR, https://www.nihr.ac.uk/) [grant code: MC_PC_19027], and Genome Research Limited, operating as the Wellcome Sanger Institute (https://www.sanger.ac.uk/about/genome-research-limited/). EV acknowledges support from the Wellcome Trust (220885/Z/20/Z) and the MRC Centre for Global Infectious Disease Analysis (reference MR/R015600/1). The funders had no role in study design, data collection and analysis, decision to publish, or preparation of the manuscript.

**Competing interests:** The authors have declared that no competing interests exist.

## Author summary

The transmission of pathogens like SARS-CoV-2 is strongly affected by chance effects in the contact process between infected and susceptible individuals, collectively referred to as random genetic drift. We have an incomplete understanding of how genetic drift changes across time and locations. To address this gap, we developed a computational method that infers the strength of genetic drift from time series genomic data that corrects for non-biological noise and is computationally scalable to the large numbers of sequences available for SARS-CoV-2, overcoming a major challenge of existing methods. Using this method, we quantified the strength of genetic drift for SARS-CoV-2 transmission in England throughout time and across locations. These estimates constrain potential mechanisms and help parameterize models of SARS-CoV-2 evolution. More generally, the computational scalability of our method will become more important as increasingly large genomic datasets become more common.

## Introduction

Random genetic drift is the change in the composition of a population over time due to the randomness of birth and death processes. In pathogen transmission, births occur as a result of transmission of the pathogen between hosts and deaths occur as a result of infected host recovery or death. The strength of genetic drift in pathogen transmission is determined by the disease prevalence, the disease epidemiology parameters [1], the variance in offspring number (the number of secondary infections that result from an infected individual) [2], as well as host contact patterns [3]. Many diseases have been found to exhibit high levels of genetic drift, such as SARS, MERS, tuberculosis, and measles [2, 4, 5]. The strength of genetic drift affects how the disease spreads through the population [2, 3, 6] how new variants emerge [7–11], and the effectiveness of interventions [12], making it an important quantity to accurately estimate for understanding disease epidemiology, evolution, and control.

The effective population size is often used to quantify the strength of genetic drift; it is the population size in an idealized Wright-Fisher model (with discrete non-overlapping generations, a constant population size, and offspring determined by sampling with replacement from the previous generation) that would reproduce the observed dynamics [13]. In a neutral population, if the effective population size is lower than the true population size, it is an indication that there are additional sources of stochasticity beyond random sampling with replacement; thus, a lower effective population size indicates a higher level of genetic drift.

Transmission of SARS-CoV-2 has been shown to exhibit high levels of superspreading (high variance in offspring number) [14–16] and high levels of genetic drift (low effective population sizes) [17–19] (see also S1 Table). However, studies have focused on particular times and locations, and we lack systematic studies over time and space (see Ref. [20] for a recent first study that uses contact tracing data to infer changes in SARS-CoV-2 superspreading over time in Hong Kong). Performing a systematic study may be most feasible with a large-scale surveillance dataset, such as that from the COVID-19 Genomics UK (COG-UK) consortium, which has sequenced almost 3 million cases of SARS-CoV-2 in both surveillance and non-surveillance capacities as of October 5, 2022. We focus specifically on this dataset, and specifically on England, due to its consistently large number of sequenced SARS-CoV-2 cases since early in the pandemic.

A challenge to performing a systematic study of the strength of genetic drift for SARS-CoV-2 and other pathogens is how to handle measurement noise, or noise from the data collection

process [21]. Measurement noise can arise from a variety of factors, including variability in the testing rate across time, geographic locations, demographic groups, and symptom status, and biases in contact tracing. Methods exist to infer measurement noise from time-series lineage or allele frequencies [22–24] (see S1 Appendix for a summary of other methods used for inferring genetic drift and additional references). Intuitively, in time-series frequency data, genetic drift leads to frequency fluctuations whose magnitudes scale with time, whereas measurement noise leads to frequency fluctuations whose magnitudes do not scale with time (Fig 1a). Thus, this system has been mapped onto a Hidden Markov Model (HMM) where the processes of genetic drift and measurement noise determine the transition and emission probabilities, respectively [25, 26]. Methods often assume uniform sampling of infected individuals from the population [22, 23, 27], but this assumption does not usually hold outside of surveillance studies. A recent study accounted for overdispersed sampling of sequences in the inference of fitness coefficients of SARS-CoV-2 variants, but assumes constant overdispersion over time [28]; in reality, the observation process may change over time due to changes in testing intensity between locations and subpopulations. Thus, to achieve the goal of systematically assessing the strength of genetic drift over time and space, there is a need to develop methods that account for time-varying overdispersed measurement noise to more accurately capture the noise generated from the observation process.

In this study, we develop a method to jointly infer genetic drift and measurement noise that allows measurement noise to be overdispersed (rather than uniform) and for the strength of overdispersion to vary over time (rather than stay constant). This method makes use of all sequencing data, which is difficult to do with existing phylogenetic methods. By fitting this model to observed lineage frequency trajectories from simulations, we show that the effective population size and the strength of measurement noise can be accurately determined in most situations, even when both quantities are varying over time. We then apply our validated method to estimate the strengths of genetic drift and measurement noise for SARS-CoV-2 in England across time (from March 2020 until December 2021) and space using over 490,000 SARS-CoV-2 genomic sequences from COG-UK. We find high levels of genetic drift for SARS-CoV-2 consistently throughout time that cannot be explained by literature values of superspreading. We discuss how community structure in the host contact network may partially explain these results. Additionally, we observe that sampling of infected individuals from the population is mostly uniform for this dataset, and we also find evidence of spatial structure in the transmission dynamics of B.1.177, Alpha, and Delta.

## Results

### Scalable method for jointly inferring genetic drift and measurement noise from time-series lineage frequency data

We first summarize the statistical inference method that we developed to infer time-varying effective population sizes from neutral lineage frequency time series that are affected by overdispersed measurement noise (more variable than uniform sampling). We explain the method more extensively in the Methods. We infer the effective population size that a well-mixed population would have to have to generate the magnitude of the fluctuations that are observed, which is the classical definition of effective population size [13]. Briefly, we use a Hidden Markov Model (HMM) with continuous hidden and observed states (a Kalman filter), where the hidden states are the true frequencies ($f_t$, where $t$ is time), and the observed states are the observed frequencies ($f_t^{obs}$) (Fig 1b) (see Methods).

The transition probability between hidden states of the HMM is set by genetic drift, where the mean true frequency is the true frequency at the previous time $E(f_{t+1}|f_t) = f_t$, and when the

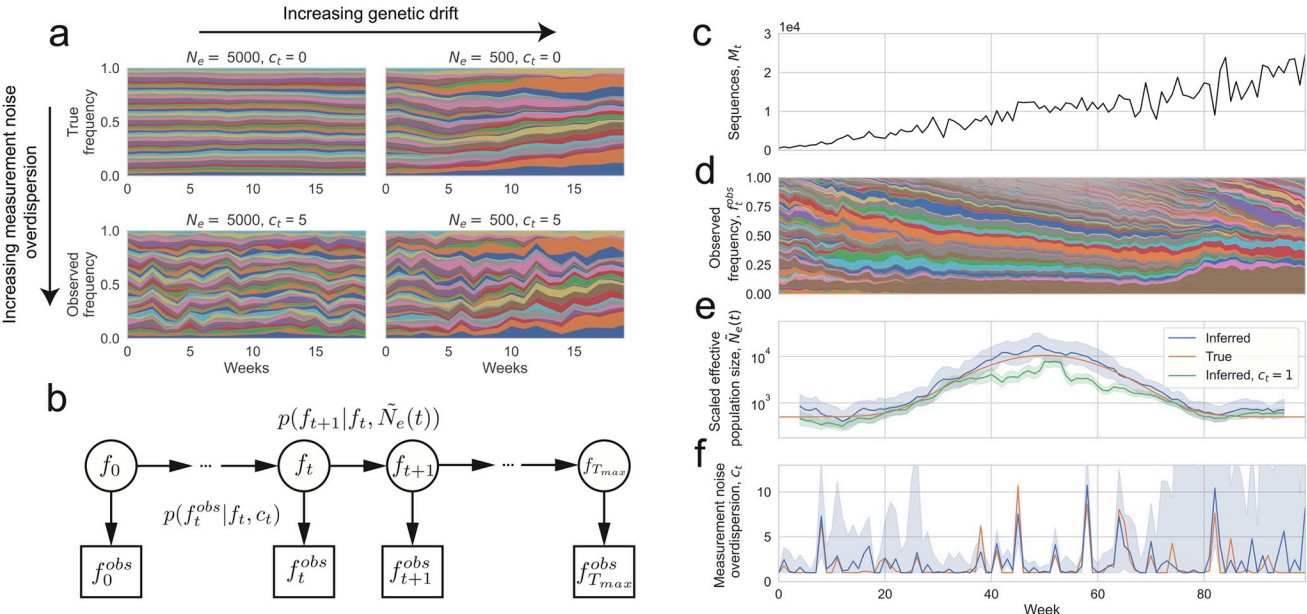

**Fig 1. A Hidden Markov Model with continuous hidden and observed states (a Kalman filter) for inferring genetic drift and measurement noise from lineage frequency time series.** (a) Illustration of how genetic drift and measurement noise affect the observed frequency time series. Muller plot of lineage frequencies from Wright-Fisher simulations with effective population size 500 and 5000, with and without measurement noise. In simulations with measurement noise, 100 sequences were sampled per week with the measurement noise overdispersion parameter $c_t = 5$ (parameter defined in text). All simulations were initialized with 50 lineages at equal frequency. A lower effective population size leads to larger frequency fluctuations whose variances add over time, whereas measurement noise leads to increased frequency fluctuations whose variances do not add over time. (b) Schematic of Hidden Markov Model describing frequency trajectories. $f_t$ is the true frequency at time $t$ (hidden states) and $f_t^{obs}$ is the observed frequency at time $t$ (observed states). The inferred parameters are $\tilde{N}_e(t) \equiv N_e(t)\tau(t)$, the effective population size scaled by the generation time, and $c_t$, the overdispersion in measurement noise ($c_t = 1$ corresponds to uniform sampling of sequences from the population). (c-f) Validation of method using Wright-Fisher simulations of frequency trajectories with time-varying effective population size and measurement noise. (c) Simulated number of sequences. (d) Simulated lineage frequency trajectories. (e) Inferred scaled effective population size ($\tilde{N}_e(t)$) on simulated data compared to true values. (f) Inferred measurement noise ($c_t$) on simulated data compared to true values. In (e) the shaded region shows the 95% confidence interval calculated using the posterior, and in (f) the shaded region shows the 95% confidence interval calculated using bootstrapping (see Methods).

frequencies are rare the variance in frequency is proportional to the mean, $\mathrm{Var}(f_{t+1}|f_t) = \frac{f_t}{\tilde{N}_e(t)}$. $\tilde{N}_e(t) = N_e(t)\tau(t)$ where $N_e(t)$ is the effective population size and $\tau(t)$ is the generation time, and both quantities can vary over time; however, we are only able to infer the compound parameter $N_e(t)\tau(t)$.

The emission probability between hidden and observed states of the HMM is set by measurement noise, where the mean observed frequency is the true frequency $\mathrm{E}(f_t^{obs}|f_t) = f_t$ and when the frequencies are rare the variance in the observed frequency is proportional to the mean, $\mathrm{Var}(f_t^{obs}|f_t) = c_t \frac{f_t}{M_t}$. $M_t$ is the number of sequences at time $t$. $c_t$ is the variance over the mean of the observed number of positive cases of each lineage at time $t$ given the true number of cases of each lineage at time $t$ (see Materials and methods). $c_t$ is expected to equal one if a random subsample of cases are sequenced, so that the observed number of cases of each lineage is approximately given by a Poisson distribution with the mean being the true number of cases of that lineage. In our analyses, we constrain $c_t \geq 1$ because realistically there must be at least Poisson sampling of cases for sequencing. Note that the constraint of $c_t \geq 1$ is still applicable when the number of sequenced cases is large as the variance already accounts for the number of sequences in the denominator. Our model assumes that the number of individuals and

frequency of a lineage is high enough such that the central limit theorem applies (at least about 20 counts or frequency of 0.01); to meet this condition, we created "coarse-grained lineages" where we randomly and exclusively grouped lineages together such that the sum of their abundances and frequencies was above this threshold (see Methods). Note that there are still sufficiently many coarse-grained lineages defined in the simulations and empirical analyses such that the assumption of the coarse-grained lineages being rare is true (needed for the defined transition and emission probabilities).

Using the transition and emission probability distributions (see Methods) and the HMM structure, we determine the likelihood function (Eq 13 in Methods) describing the probability of observing a particular set of lineage frequency time-series data given the unknown parameters, namely the scaled effective population size across time $\tilde{N}_e(t)$ and the strength of measurement noise across time $c_t$. We then maximize the likelihood over the parameters to determine the most likely parameters that describe the data. Because we are relying on a time-series signature in the data for the inference, we need to use a sufficiently large number of timesteps of data; on the other hand, the longer the time series, the more parameters would need to be inferred (since both $\tilde{N}_e(t)$ and $c_t$ are allowed to change over time). To balance these two factors, we assumed that the effective population size stays constant over a time period of 9 weeks (a form of "regularization"). We then shift this window of 9 weeks across time to determine how $\tilde{N}_e(t)$ changes over time (see Methods), but this effectively averages the inferred $\tilde{N}_e(t)$ over time. $c_t$ is still allowed to vary weekly.

To validate our model, we ran Wright-Fisher simulations with time-varying effective population size and time-varying measurement noise (Fig 1c–1f). Because a substantial number of lineages would go extinct over the simulation timescale of 100 weeks, we introduced new lineages with a small rate (a rate of 0.01 per week per individual of starting a new lineage) to prevent the number of lineages from becoming too low. We then did inference on the simulated time-series frequency trajectories (Fig 1d). The inferred $\tilde{N}_e(t)$ and $c_t$ closely follow the true values (Fig 1e–1f), and the 95% confidence intervals (see Methods for how they are calculated) include the true value in a median (across timepoints) of 95% of simulation realizations (S1 Fig). The error in $c_t$ is higher when the variance contributed to the frequency trajectories by measurement noise is lower than that of genetic drift, which occurs when the effective population size is low or number of sequences is high (more clearly seen in S2 Fig, where the effective population size is held constant). However, the error on $\tilde{N}_e(t)$ seems to be unchanged or even slightly decrease when the error on $c_t$ is increased because the contribution to the variance due to genetic drift is higher. We also observe that the inferred $\tilde{N}_e(t)$ is smoothed over time due to the assumption of constant $\tilde{N}_e(t)$ over 9 weeks (S3 Fig); this is a potential drawback when there are sharp changes in the effective population size over time. Importantly, we observed that the inferred $\tilde{N}_e(t)$ will be underestimated if sampling is assumed to be uniform when it is actually overdispersed (Fig 1e). This is because variance in the frequency trajectories due to measurement noise is incorrectly being attributed to genetic drift. The underestimation is strongest when the variance contributed due to measurement noise is high, either due to high measurement noise overdispersion, a low number of sampled sequences, or a high effective population size. In this situation, joint inference of measurement noise and $\tilde{N}_e(t)$ from the data is necessary for accurate inference of $\tilde{N}_e(t)$.

In summary, we developed a method to infer the strength of genetic drift and measurement noise from lineage frequency time series data and validated the accuracy of the method with simulations. This method has the potential to scale well with large amounts of genomic data as it only relies on lineage frequency time series data.

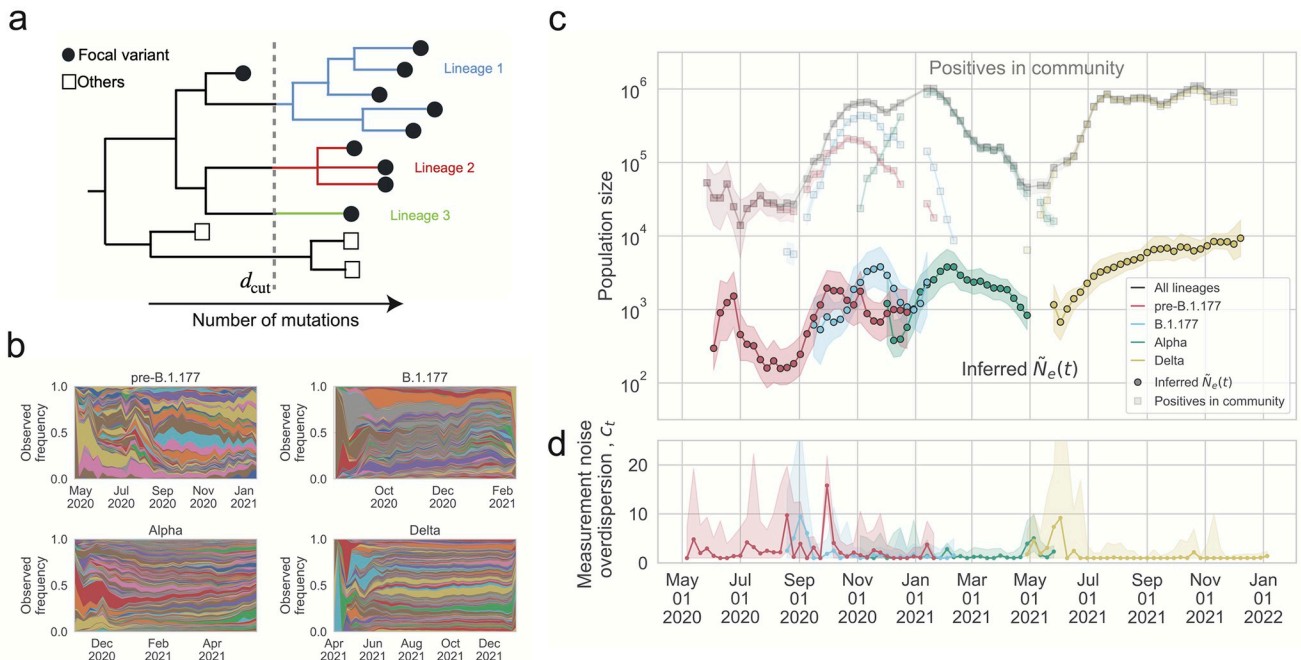

**Fig 2. The inferred effective population size and overdispersion of measurement noise in England compared with the number of positive individuals.**
(a) Schematic of lineage construction for B.1.177, Alpha, and Delta from the COG-UK phylogenetic tree. The filled circles represent the sequences of a focal variant sampled in England, while the unfilled squares represent other sequences, which are of other variants or sampled in other countries. The phylogenetic tree is cut at a certain depth $d = d_{cut}$, and each branch cut by the line $d = d_{cut}$ defines a lineage. Lineages pre-B.1.1.7 are defined using the pango nomenclature. (b) Muller plot of lineage frequency time series for lineages pre-B.1.177, of B.1.177, of Alpha, of Delta. (c) Inferred scaled effective population size ($\tilde{N}_e(t) \equiv N_e(t)\tau(t)$) for pre-B.1.177 sequences, B.1.177, Alpha, and Delta, compared to the estimated number of people testing positive for SARS-CoV-2 in England at the community level, as measured by the COVID-19 Infection Survey, for all lineages and by variant or group of lineages. To simplify the plot, only data where the number of positive individuals for a given variant or group of lineages was higher than $10^3$ in a week are shown. The inferred $\tilde{N}_e(t)$ is considerably lower than the number of positive individuals for all times and for all variants or group of lineages. (d) Inferred measurement noise overdispersion ($c_t$) for pre-B.1.177 sequences, B.1.177, Alpha, and Delta.

### Inference of genetic drift in SARS-CoV-2 transmission in England

We next applied this method to study the effective population size and strength of measurement noise for SARS-CoV-2 in England, where hundreds of thousands of SARS-CoV-2 genomes have been sequenced. Because our method assumes that lineages are neutral with respect to one another (no selection), we performed separate analyses on groups of lineages that have been shown to exhibit fitness differences or deterministic changes in frequency: lineages pre-B.1.177, B.1.177, Alpha, and Delta [17, 28–30]. We checked that the assumption of neutrality within each of these groups does not significantly affect our results, and this is described below.

To obtain lineage frequency time series data for SARS-CoV-2 in England, we downloaded genomic metadata from the COVID-19 Genomics UK Consortium (COG-UK) [31] (Fig 2b) and the associated phylogenetic trees that were created at different points in time. To minimize potential bias, we used only surveillance data (labeled as "pillar 2"). For sequences pre-B.1.177, we used the pangolin lineages assignments from COG-UK [32, 33]. However, B.1.177, Alpha, and Delta were subdivided into only one or a few pangolin lineages, since a new lineage is defined by sufficiently many mutations and evidence of geographic importation. However, for our purposes we only need resolution of neutral lineages within a variant. Thus, we created additional neutral lineages by cutting the phylogenetic tree at a particular depth and grouping

sequences downstream of the branch together into a lineage (see Fig 2a and Methods). Note that as a result, the "lineages" that we define here are not necessarily the same as the lineages defined by the Pango nomenclature. The trees were created by COG-UK and most sequenced samples were included in the trees (S4 Fig). However, in some instances downsampling was necessary when the number of sequences was very large. In these situations, any downsampling (performed by COG-UK) was done by trying to preserve genetic diversity. Most sequences in the tree were assigned to lineages (see Methods), and we corrected for the fraction of sequences that were not assigned to lineages in our inference of $\tilde{N}_e(t)$ (see Methods). This yielded 486 lineages for pre-B.1.177, 4083 lineages for B.1.177, 6225 lineages for Alpha, 24867 lineages for Delta.

The inferred scaled effective population size ($\tilde{N}_e = N_e \tau$, effective population size times generation time, where the generation time is the time between infections in infector-infectee pairs) is shown in Fig 2c. The generation time is around 4–6 days (0.6–0.9 weeks) depending on the variant [34, 35], but we leave the results in terms of the scaled effective population size (rather than effective population size) because the generation time may change over time [34], has a high standard deviation [34], and is close to one week so is expected to not drastically change the result; additionally, as we show below, the null model estimate that we compare to is also multiplied by the generation time, which cancels when we look at the ratio (described below). The scaled inferred effective population size was lower than the number of positive individuals in the community (estimated by surveillance testing from the COVID-19 Infection Survey [36] and see Methods) by a factor of 20 to 1060 at different points in time. The most notable differences between the changes over time in the number of positives in the community and that of the scaled effective population size were: the inferred scaled effective population size of lineages pre-B.1.177 peaked slightly before the number of pre-B.1.177 positives peaked, the inferred scaled effective population size of Alpha decreased slower than the number of positives decreased after January 2021, and the shoulder for the inferred scaled effective population size of Delta occurred earlier than in the number of positives. We checked that the inferred scaled effective population size is not sensitive to the depth at which the trees are cut to create lineages (S5, S6 and S7 Figs), the threshold counts for creating coarse-grained lineages (S8 Fig), or the number of weeks in the moving time window (S9 Fig). Additionally, we checked that the gaussian form of the transition and emission probabilities in the HMM are a good fit to the data (S10 Fig).

The inferred measurement noise for each group of lineages is shown in Fig 2d. The inferred measurement noise overdispersion was mostly indistinguishable from 1 (uniform sampling), but at times was above 1 (sampling that is more variable than uniform sampling). There were also at times differences in the strength of measurement noise between variants when they overlapped in time. In particular, measurement noise for lineages pre-B.1.177 peaked in October 2020 despite measurement noise being low for B.1.177 at that time.

To better interpret the observed levels of genetic drift, we compared the inferred $\tilde{N}_e(t)$ to that of an SIR null model, which includes a susceptible, infectious, and recovered class. The $\tilde{N}_e(t)$ for an SIR model was derived in Ref. [37–39] and is given by

$$\tilde{N}_e^{\text{SIR}}(t) = \frac{I(t)}{2R_t \gamma_I} \tag{1}$$

where $I(t)$ is number of infectious individuals, $R_t$ is the effective reproduction number, and $\gamma_I$ is the rate at which infectious individuals recover. For the number of infectious individuals, we used the number of positive individuals estimated from the UK Office for National Statistics' COVID-19 Infection Survey [36], which is a household surveillance study that reports positive

PCR tests, regardless of symptom status. We used the measured effective reproduction number in England reported by the UK Health Security Agency [40]. We used $\gamma_I^{-1} = 5.5$ days [41, 42], and our results are robust to varying $\gamma_I$ within a realistic range of values (S11 Fig). We found that $\tilde{N}_e^{\mathrm{SIR}}(t)$ is very similar to the number of positives because the effective reproduction number in England was very close to 1 across time and $\gamma_I$ is also very close to 1 in units of weeks$^{-1}$. To calculate $\tilde{N}_e^{\mathrm{SIR}}(t)$ for each variant or group of lineages, we rescaled the population-level $I(t)$ and $R_t$ based on the fraction of each variant in the population and the relative differences in reproduction numbers between variants (see Methods). We then calculated the scaled true population size, $\tilde{N}(t) \equiv N(t)\tau(t)$, for the SIR model by multiplying by the variance in offspring number, $\sigma^2$, for the SIR model [43]

$$\tilde{N}^{\mathrm{SIR}}(t) = \tilde{N}_e^{\mathrm{SIR}}(t)\{\sigma^2\}^{\mathrm{SIR}} \tag{2}$$

$$\{\sigma^2\}^{\mathrm{SIR}} = 2. \tag{3}$$

Overall, the inferred $\tilde{N}_e(t)$ is lower than $\tilde{N}^{\mathrm{SIR}}(t)$ by a time-dependent factor that varies between 20 and 590 (Fig 3c and S12 Fig), suggesting high levels of genetic drift in England across time. We find similar results when using an SEIR rather than an SIR model which additionally includes an exposed class and may be more realistic (Methods, S1 Appendix, and S13 Fig). The ratio of $\tilde{N}^{\mathrm{SIR}}(t)$ to the inferred $\tilde{N}_e(t)$ was similar across variants and across time, except that for Alpha the ratio initially peaked and then decreased over time.

Because non-neutral lineages could potentially bias the inferred effective population size to be lower in a model that assumes all lineages are neutral, we checked the assumption that lineages are neutral with respect to one another within a group or variant (pre-B.1.177, B.1.177, Alpha, and Delta) by detecting deterministic changes in lineage frequency. We used a conservative, deterministic method that ignores genetic drift, which is expected to overestimate the number of non-neutral lineages. We found that 50% of lineages had absolute fitness above 0.09 (above the 50th percentile) and 10% of lineages had absolute fitness above 0.27 (above the 90th percentile). Very likely, some of these lineages are detected as having non-zero fitness simply because the model does not correctly account for strong genetic drift which would also lead to changes in lineage frequency. Excluding non-neutral lineages with absolute fitness values above the 50th ($|s| > 0.09$), 75th ($|s| > 0.16$), and 90th ($|s| > 0.27$) percentiles, leads to only slight changes in the inferred effective population size (S14 Fig). This result shows that conservatively excluding lineages that could be non-neutral does not change the result that the inferred effective population size is one to two order of magnitudes lower than the SIR or SEIR model effective population size.

We also tested whether background selection (selection against deleterious mutants) in SARS-CoV-2 could be responsible for a substantial fraction of the reduction in effective population size. We simulated the lineage frequency dynamics using the empirically estimated distribution of deleterious fitness effects from Ref. [44] (S15 Fig and Methods) and found that the inferred effective population size is consistent with the true effective population size to within the error bars (S16 Fig) and lower than the inferred effective population size in a simulation with only neutral mutations (S17 Fig) by no more than a factor of 2 (S18 Fig). Analytical estimates for the expected decrease in effective population size due to the empirical distribution of deleterious fitness effects also predict at most a factor of at most 2 decrease in effective population size that is not sufficient to explain the two orders of magnitude lower effective population size that we observe compared to the expectation (S1 Appendix).

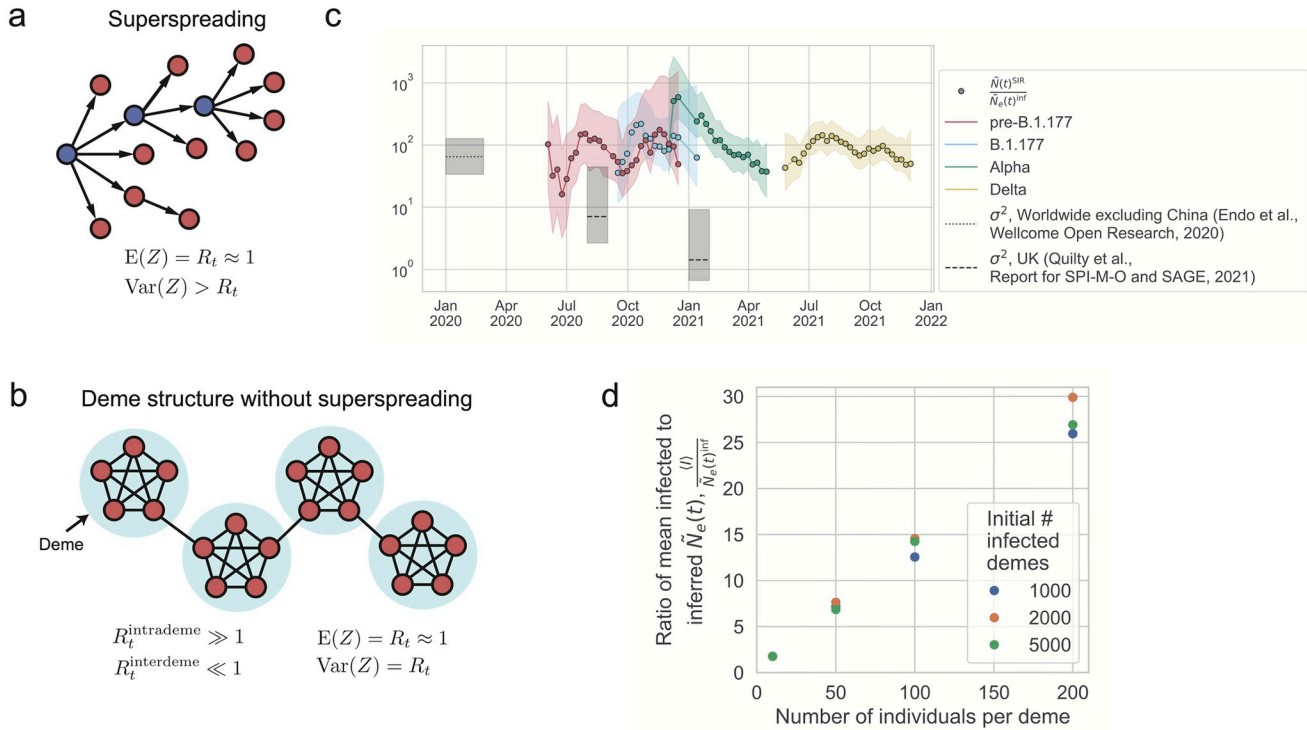

**Fig 3. Potential mechanisms that can generate a low effective population size.** (a) Superspreading, where the distribution of the number of secondary cases ($Z$) from a single infected individual is broadly distributed (variance greater than mean). The superspreading individuals are indicated in blue. (b) Deme structure without superspreading, due to heterogeneity in the host network structure, where the distribution of the number of secondary cases is not broadly distributed (variance approximately equal to mean). (c) The ratio between the $\tilde{N}^{SIR}(t)$ (the scaled population size calculated from an SIR model using the number of observed positive individuals and the observed effective reproduction number) and the inferred $\tilde{N}_e(t)$ for each variant. Only data where the error in the SIR model $\tilde{N}^{SIR}(t)$ is less than 3 times the value are shown, because larger error bars make it challenging to interpret the results. The inferred $\tilde{N}_e(t)$ is lower than the $\tilde{N}^{SIR}(t)$ (which assumes well-mixed dynamics and no superspreading) by a factor of 16 to 589, indicating high levels of genetic drift. The variance in offspring number from the literature does not entirely explain the discrepancy between the true and effective population sizes. (d) Simulations of deme structure without superspreading can generate high levels of genetic drift via jackpot events. SEIR dynamics are simulated within demes (with $R_t = 10$, i.e. deterministic transmission) and Poisson transmission is simulated between demes ($R_t \ll 1$, i.e. stochastic transmission) such that the population $R_t \sim 1$ (see Methods). Simulation parameters are: mean transition rate from exposed to infected $\gamma_E = (2.5 \text{ days})^{-1}$, mean transition rate from infected to recovered $\gamma_I = (6.5 \text{ days})^{-1}$, total number of demes $D_{total} = 5.6 \times 10^5$. The ratio between the number of infected individuals and the inferred effective population size is found to scale linearly with the deme size and not with the number of infected demes. This scaling results because of jackpot events where a lineage that happens to infect a susceptible deme grows rapidly until all susceptible individuals in the deme are infected.

We also probed the spatial structure of transmission by inferring the scaled effective population size separately for each region within England. We find that the scaled effective population size in the regions of England is substantially smaller than that in England as a whole for B.1.177, Alpha, and Delta (S19 Fig), suggesting that the transmission was not well-mixed at that time. Additionally, the discrepancy between the inferred regional scaled effective population size and the observed number of positive individuals in a region was comparable to that seen in England as a whole (S20 Fig), which is consistent with spatially segregated dynamics with similar levels of genetic drift in each region. We further describe these results in the S1 Appendix.

## Discussion

Here, we systematically studied the strength of genetic drift of SARS-CoV-2 in England across time and spatial scales. To do this, we developed and validated a method for jointly inferring

time-varying genetic drift and overdispersed measurement noise using lineage frequency time series data (Fig 1), allowing these two effects to be disentangled, which overcomes a major challenge in the ability to infer the strength of genetic drift from time-series data. Additionally, this method makes use of all sequencing data, overcoming the need to subsample data, which is a challenge with current phylogenetic methods. Our approach was able to reproduce the expected decrease in effective population size during the decline of pre-B.1.177, B.1.177, and Alpha, as well as the increase in effective population size during the emergence of B.1.177, Alpha, and Delta (Fig 2c). We did not have enough sequences during the time when Delta was going extinct to infer the effective population size during that time period.

We find that the effective population size of SARS-CoV-2 in England was lower than that of an SIR null model true population size (using the observed number of positives) by a time-dependent factor ranging from 20 to 590 (Fig 3c), suggesting that there were higher levels of genetic drift than expected from uniform transmission. We also find evidence for spatial structure in the transmission dynamics during the B.1.177, Alpha, and Delta waves, as the inferred $\tilde{N}_e(t)$ was substantially lower in regions compared to that of all England (S19 Fig). These findings are consistent with other studies that have found spatial structure in transmission of B.1.177 [45], Alpha [46], and Delta [47].

We observed that with a few exceptions, the amount by which genetic drift was elevated compared to the number of positives did not change substantially over time or across variants outside the range of the error bars (Fig 3c), despite changes in lockdowns and restrictions (which we may expect to decrease behavior that leads to superspreading). This may be due to not having enough statistical power due to the dataset size. On the other hand, we note that restrictions affect the mobility network structure in a complex way, decreasing some types of mobility while increasing others [48], so lockdowns and restrictions may not affect the effective population size in a predictable way. One exception was that Alpha had significantly higher genetic drift compared to Delta and the strength of genetic drift in Alpha first peaked then slowly decreased over time. This may be either due to differences in the properties of the virus or differences in host behavior. For instance, it may suggest that the stochasticity in the transmission of Alpha sharply increased then slowly decreased over time. Alternatively, this may be driven by Alpha's expanding geographic range combined with reimported cases of Alpha into the UK (observed from February 2021 onwards), which could both also decrease the effective population size [49].

It is important to distinguish measurement noise from genetic drift as measurement noise is a function of the observation process and will not affect disease spread, extinction, and establishment of new mutations. We observe that measurement noise of SARS-CoV-2 is mostly indistinguishable from uniform sampling, but data from some variants at some times do exhibit more elevated measurement noise than uniform sampling. Thus, we expect that assuming uniform sampling, as many methods do, or constant overdispersion will lead to accurate estimates for this dataset [22, 23, 27, 28]. The number of SARS-CoV-2 sequences from England is extremely high and sampling biases are expected to be low, because of efforts to reduce sampling biases by sampling somewhat uniformly from the population through the COVID-19 Infection Survey [36] (from which a subset of positives are sequenced and included in the COG-UK surveillance sequencing data that we use). On the other hand, other countries may have higher sampling biases, so jointly estimating measurement noise and genetic drift may be more crucial in those settings. It may also be interesting to use this method to test whether genomics data taken from wastewater has lower levels of measurement noise as compared to sequenced cases.

We find that constant selection is unlikely to explain our results, as liberally excluding potentially non-neutral lineages does not significantly change the inferred effective population size. Our method is not able to precisely pinpoint how many lineages are under selection, but it appears that there is relatively little within-variant selection in the time period we investigated, and our method is robust to slight deviations from neutrality. Additionally, background selection is unlikely to explain our results as the empirically estimated distribution of deleterious fitness effects for SARS-CoV-2 decreased the effective population size by at most a factor of 2 from that of the completely neutral scenario.

Accurately estimating the strength of genetic drift allows us to better understand disease spread and extinction, as well as to better parameterize evolutionary models and understand how mutations will establish in the population. The establishment probability is the probability that a new mutation will rise to a high enough frequency to escape stochastic extinction. For weakly beneficial mutations, the establishment probability is linearly related to the effective population size [50]. For strongly beneficial mutations, the impact of the effective population size on the establishment probability is quantitatively less straightforward and depends on the host network structure [3]. In the absence of clonal interference, the fixation probability, or the probability that the mutation will fix in a population, is the same as the establishment probability; if there is clonal interference, the fixation probability will depend on additional factors like the mutation rate [51, 52]. The low effective population sizes that we observe suggest low establishment probabilities; the probability that any newly arisen beneficial mutant rises to a significant frequency will be small. More generally, our results give an order of magnitude estimate for the effective population sizes that can be used to more accurately parameterize evolutionary models for SARS-CoV-2 as well as an approach to infer the effective population size in more specific contexts.

## Potential mechanisms that can contribute to the high levels of genetic drift

Two potential mechanisms that can contribute to the observed high levels of genetic drift are: (1) variability at the individual level through superspreading (Fig 3a), and (2) host population structure (Fig 3b). We investigate each of these mechanisms in turn and compare it to our results. While in reality, both mechanisms (and others not explored here) are likely at play, it is challenging to tease them apart given our limited data. Therefore, in order to gain intuition about how each of these phenomena drives the strength of genetic drift in this system, we consider each in turn.

Infected individuals that cause an exceptional numbers of secondary cases (superspreaders) are one reason for an increased level of allele frequency fluctuations. The expected decrease in effective population size is given by the per-generation variance in secondary cases, which is sensitive to superspreaders broadening the tail of the offspring distribution. Direct measurements of the offspring distribution through contact tracing yield variances substantially smaller than our inferred reduction in effective population size [53–57] (S1 Table and Fig 3c). This could indicate that the tail of the offspring distribution is not well measured by contact tracing efforts or that other factors are at play that could decrease the effective population size.

Primary factors that could further increase fluctuations are selection and spatial structure. While both positive and background selection have some effect, we estimate their contribution to not exceed a factor of 2 (see above and S1 Appendix). We now show that, by contrast, a pronounced host deme structure can easily decrease the effective population size by orders of magnitude, even without individual super spreaders.

Consider a model in which individuals within a deme are very well-connected to one another (i.e. households or friend groups, also known as "communities" in network science

[58]), but there are few connections between demes (Fig 3b). It is possible for deme structure to occur without superspreading. Because individuals are very well-connected within a deme, once the pathogen spreads to a susceptible deme, it will spread rapidly in a deme until all individuals are infected (a jackpot event). In this way, deme structure can lower the effective population size by lowering the effective number of stochastic transmissions events. For instance, in the example in Fig 3b, there are 20 individuals, but only 3 potential stochastic transmissions events. Deme structure may also arise from correlations in the number of secondary infections over a series of hosts (i.e. a series of high numbers of secondary infections in a transmission chain, or conversely low numbers of secondary infections in a transmission chain) [59]. This may arise, for instance, if individuals in a transmission chain have similar behavior, due to geographical proximity, or similar value systems on risk aversion. A recent study has found that individuals infected by superspreading tend to be superspreaders themselves more often than expected by chance [60], which would be consistent with this phenomenon.

To check our intuition that deme structure can decrease the effective population size and increase genetic drift, we ran simulations of a simplified deme model (see Methods): all demes have the same number of individuals, and there is a sufficiently large enough number of demes that the total number of demes does not matter. Initially a certain number of demes are infected, and transmission occurs such that the overall effective reproduction number in the population is around 1. From our simulations, we find that when the number of individuals in a deme increases, the ratio between the number of infected individuals and the inferred effective population size increases (Fig 3d); in other words, the more individuals there are in a deme, the higher the level of genetic drift we observe compared what is expected from the number of infected individuals. This is because while the number of infected individuals increases when the deme size increases (S21a Fig), the inferred effective population size (and thus the level of stochasticity) stays the same as a function of deme size (it is more dependent on the number of infected demes) (S21b Fig). However, the exact ratio of the number of infected individuals to the inferred effective size depends on the parameters of the model.

Studies that inferred the overdispersion parameter for the offspring number distribution using modeling rather than direct contact tracing and found a high variance in offspring number (see S1 Table; for example, Ref. [61]) may actually be consistent with our results as the high variance may be partly due to superspreading events from, for example, host deme structure.

In reality, both superspreading and host structure are likely at play. Additionally, they could interact with each other. For instance, there could be superspreading within a deme. Future work can try to tease apart the contribution of these two mechanisms, which for instance may be possible with better transmission network data, building on previous work on transmission networks [62], or with time-resolved contact tracing data [20]. This will be important because the relative contributions of the two mechanisms of superspreading and host population structure to genetic drift can affect the establishment of new variants in the population in different ways [3]. If our interpretation is correct that deme structure and jackpot events strongly affect the effective population size, then managing superspreading events will be important to decrease the strength of genetic drift; nonpharmaceutical interventions should try to reduce these types of events.

## Limitations of the study and opportunities for future directions

First, the quantity of effective population size is a summary statistic that is influenced by many factors, making its interpretation challenging. The effective population size describes the population size under a well-mixed Wright-Fisher model, whereas in reality, this assumption is broken by selection, migration, host structure, broad offspring number distributions, mutation,

within-host evolution, and many other evolutionary and demographic processes. While many of these processes jointly contribute to the strength of genetic drift at the transmission level (broad offspring number distributions, host structure), which is what we are interested in inferring in this study, some other processes may confound the inference of genetic drift at the transmission level (selection, migration, within-host evolution, etc). While it would have been computationally intractable to jointly infer all possible processes, we addressed the processes that we thought were most likely to affect the effective population size in this system besides genetic drift at the transmission level.

We checked that constant selection could not lower the effective population size as much as we observed. We did not test for more complex forms of selection, such as fluctuating selection, because including more complex forms of selection quickly increases the number of parameters in the model such that it becomes intractable. However, we note that fluctuating selection that occurs on a fast enough time scale will act effectively like genetic drift in increasing stochasticity in transmission. We ignored importation of SARS-CoV-2 into England and exportation of SARS-CoV-2 out of England. Migration can substantially change frequencies that are locally rare, but we expect importations to only weakly influence the frequency fluctuations of abundant variants, on which we have focused in this work. Host migration within the population can lead to gene flow; however, this will only affect the effective population size if it results in jackpot events [13]. Our model of host deme structure does indeed incorporate gene flow within the population with jackpot events, and we find that this type of host deme structure can substantially decrease the effective population size.

Empirically measured SARS-CoV-2 offspring distributions that take into account superspreaders (see references in S1 Table) have been described by a negative binomial distribution, which has a finite mean and variance and thus can be described by the Wright-Fisher model. We focused on standing variation that existed at a particular depth in the phylogenetic tree and ignored de novo mutations subsequently arising during the time series. However, we don't think this should substantially affect our results because introducing mutations in the form of new lineages with a small rate in the simulations did not have a large effect on the method performance (Fig 1e). While within-host dynamics may in principle impact the lineage frequency trajectories, this effect is likely small for our analysis because we focus on acute infections (infections in the community rather than in hospitals and nursing homes). Acute infections of SARS-CoV-2 are thought to generate little within-host diversity that is passed on due to the short infection duration and small bottleneck size between hosts [63, 64]; while new mutations arising within acute hosts have been observed to be transmitted, these events are rare [63].

Thus, we think to the best of our knowledge that the low effective population sizes that we observe are due to increased levels of genetic drift at the transmission level, which can be due to a variety of mechanisms, including the two that we highlight above, superspreading and host deme structure. However, future work should explore joint inference of selection, migration, and/or mutation in the model, as is appropriate for the pathogen of interest, building on previous work in this area [26, 65–67].

Second, there may be biases in the way that data are collected that are not captured in our model. While our method does account for sampling biases that are uncorrelated in time, sampling biases that remain over time cannot be identified as such (i.e. if one geographical region was dominated by a particular lineage and it consistently had higher sequencing rates compared to another geographical region), and this can potentially bias the inferred effective population size; although, this is also a problem in phylogenetic methods. One approach to this problem that was utilized by some early methods during the pandemic is to develop sample weights based on geography, time, and number of reported cases. Future work should study

the effect of different sampling intensities between regions on uncorrelated and correlated sampling noise. Additionally, we assume that the measurement noise overdispersion is identical for all lineages within a variant; in reality, there may be differences in sampling between lineages. However, we do not expect this to have a large effect on our results as we observed that measurement noise overdispersion was close to 1 for most timepoints in this dataset. Future work can test the effect of lineage-specific measurement noise overdispersion on overall method performance across different datasets.

Third, the use of a sliding window of 9 weeks on the lineage frequency data will lead to smoothing of sharp changes in effective population size. In our analysis, shortening the time window did not substantially affect our results. It may be interesting in future work to develop a continuous method that uses a prior to condition on changes in effective population size, similar to those that have been developed for coalescence-based methods [1, 68]. This would allow us to infer continuous changes in effective population size without needing to use a sliding window.

Fourth, we have defined lineages by cutting the phylogenetic tree at a particular depth; we chose this approach because a tree available for these sequences from COG-UK and we wanted to be somewhat consistent with the existing pango nomenclature for SARS-CoV-2 lineages, which were defined using a tree. One concern is that errors in the constructed tree may introduce additional fluctuations to the lineage frequencies. This may particularly be a problem for SARS-CoV-2 given the low mutation rate. As one check, we tested that cutting the tree at different depths did not affect the results (S5 Fig), suggesting that our results were not sensitive to differences in lineage definitions at those depths. However, lineages defined using the two cut depths may both have errors in the groupings, so to be more robust, future work could systematically investigate the sensitivity of our method to errors in the tree or compare the results using lineage frequencies and allele frequencies (defined using mutations). Recent advances have made building trees for large datasets more tractable [69], but we can potentially increase the scalability of our approach even further by making the method tree-free. For example, one idea is to cluster the sequences based on a distance metric and use cluster frequencies over time or another idea is to use allele frequencies (the frequencies of individual mutations). Future work should evaluate the feasibility and accuracy of using these different approaches to process the data for inferring the effective population size.

While we have focused on SARS-CoV-2 in this study, our simulations point to the generalizability of our approach, and the method developed here can be extended to study genetic drift in other natural populations that are influenced by measurement noise and where genomic frequency data are available. We think that this approach would be best suited for large datasets with a long period of sampling, and for pathogens this includes HIV, Ebola, and potentially seasonal influenza. It may also be interesting to adapt this approach to study data from field studies and ancient DNA [70–72]. More generally, ongoing methods development that integrates genomics, epidemiological, and other data sources is crucial for being able to harness the large amounts of data that have been generated to better understand and predict evolutionary dynamics.

## Materials and methods

### Data sources and processing

We downloaded sequence data from the COVID-19 Genomics UK Consortium (COG-UK) [31]. We only used surveillance data (labeled as "pillar 2"); this dataset is composed of a random sample of the positive cases from the COVID-19 Infection Survey, which is a surveillance study of positive individuals in the community administered by the Office for National

Statistics (see below). For lineages that appeared before B.1.177, we downloaded the metadata from the COG-UK Microreact dashboard [73], which included the time and location of sample collection (at the UTLA level), as well as the lineage designation using the Pango nomenclature [32, 33]. For B.1.177, Alpha, and Delta sequences, because the Pango nomenclature classified them into very few lineages, we created our own lineages from the phylogenetic trees (see below). We downloaded the publicly available COG-UK tree on February 22, 2021 for B.1.177; June 20, 2021 for Alpha; and January 25, 2022 for Delta. Additional sensitively analyses shown in S5, S6 and S7 Figs used trees downloaded on June 1, 2021 for Alpha and March 25, 2022 for Delta. The publicly available trees were created by separating sequences into known clades, running fasttree [74] separately for each clade, grafting together the trees of different clades, and then using usher [69] to add missing samples (code available at https://github.com/virus-evolution/phylopipe). We also downloaded the COG-UK metadata for all lineages on January 16, 2022, which included the time and location (at the UTLA level) of sample collection. Additional sensitivity analyses shown in S6 and S7 Figs used metadata downloaded on March 25, 2022. For the data of B.1.177, Alpha, and Delta, the data was deduplicated to remove reinfections in the same individual by the same lineage, but reinfections in the same individual by a different lineage were allowed. This yielded a total of 490,291 sequences.

The lineage frequency time-series is calculated separately for each variant or group of lineages (pre-B.1.177, B.1.177, Alpha, and Delta). First, the sequence metadata are aggregated by epidemiological week (Epiweek) to average out measurement noise that may arise due to variations in reporting within a week. Then, the lineage frequency is calculated by dividing the number of sequences from that lineage in the respective tree by the total number of sequences of that variant (or group of lineages) that were assigned to any lineage in the respective tree.

Because our model describes birth-death processes when the central limit theorem can be applied, we need the lineage frequencies to be sufficiently high. Thus, we randomly combine rare lineages into "coarse-grained lineages" that are above a threshold number of counts and threshold frequency in the first and last timepoint of each trajectory. The motivation of having a cutoff for both counts and frequency is to account for the fact that the total number of counts (number of sequences) varies over time. For the threshold, we chose 20 counts and frequency of 0.01. The motivation for combining lineages together randomly was to further remove any potential effects due to selection. We also tested that creating lineages by cutting the tree closer to the root of the tree did not substantially affect the results (S5 and S6 Figs); this shows that grouping lineages together based on genetic similarilty would not have had a substantial affect on our results. Sensitivity analyses showed that the choice of the coarse-grained lineage count threshold does not substantially affect the results (S8 Fig). Coarse-grained lineages are non-overlapping (i.e. each sequence belongs to exactly one coarse-grained lineage).

The estimated number of people testing positive for COVID-19 in England and each region of England was downloaded from the UK Office for National Statistics' COVID-19 Infection Survey [36]. The COVID-19 Infection Survey includes households that are semi-randomly chosen, and individuals are tested regardless of whether they are reporting symptoms. Infections reported in hospitals, care homes, and other communal establishments are excluded. Thus the dataset provides a representative number of positive individuals in the community setting. The reported date of positive cases is the date that the sample was taken. The error on the number of positive individuals from April 17, 2020 to July 5, 2020 is reported as the 95% confidence interval, and after July 5, 2020 is reported as the 95% credible interval. The regional data reported the positivity rate over two week intervals. To get the number of positives, we multiplied by the number of individuals in the community setting in the region (excluding hospitals, care homes, and other communal establishments). As the data was reported over

two week intervals, we obtained the number of positives for each week using linear interpolation.

The observed effective reproduction numbers for England and each region of England were downloaded from the UK Health Security Agency [40]. Only times where the certainty criteria are met and the inference is not based on fewer days or lower quality data are kept. The error on the effective reproduction number is reported as the 90% confidence interval. Although not reported in the dataset, we choose the point estimate of the effective reproduction number to be the midpoint between the upper and lower bounds of the 90% confidence interval.

## Creating lineages in B.1.177, Alpha, and Delta

For B.1.177, Alpha, and Delta, we divided each of them into neutral lineages based on phylogenetic distance. Specifically, for B.1.177 and Alpha, we cut a phylogenetic tree (in units of number of mutations from the root of the tree) at a certain depth, $d = d_{cut}$. Each of the internal or external branches that are cut by the line $d = d_{cut}$ defines a lineage (Fig 2a). The (observed) frequency of a lineage at a given time point in England was computed by counting the number of England sequences (leaf nodes) belonging to the lineage and by normalizing it by the total number of sequences in all assigned lineages of the focal variant in England at that time point. Lineage frequencies at the regional level were similarly computed by counting the number of sequences separately for each region.

The choice of $d_{cut}$ is arbitrary to some extent. Because we wanted a sufficiently high resolution of lineages from the early phase of spreading of a variant and because the evolutionary distance correlates with the actual sample date (S22 Fig), for each focal variant, we chose the depth $d_{cut}$ that roughly corresponds to the time point when it began to spread over England.

For the Delta variant, the sequences form two distinct groups along the depth direction, as seen from the last panel of S22 Fig. Therefore, to divide the Delta variant into lineages with small frequencies, we cut the phylogenetic tree at two depths sequentially; we first cut the tree at $d_{cut}^{(1)}$, which resulted in lineages with small frequencies plus a lineage with $\mathcal{O}(1)$ frequency. Then, to divide the latter lineage further, we took the subtree associated with this lineage and cut the subtree at $d_{cut}^{(2)}$.

For the results presented in the main text, we used (in units of substitutions per site, with the reference d = 0 being the most recent common ancestor) $d_{cut} = 2.323 \cdot 10^{-2}$ for B.1.177, $d_{cut} = 2.054 \cdot 10^{-3}$ for Alpha, and $d_{cut}^{(1)} = 1.687 \cdot 10^{-3}$ and $d_{cut}^{(2)} = 1.954 \cdot 10^{-3}$ for Delta. We confirmed that our results are robust to the choice of $d_{cut}$ as well as the choice of the phylogenetic tree data we used (S5, S6 and S7 Figs).

## Model for inferring effective population size from lineage frequency time series

We use a Hidden Markov Model with continuous hidden and observed states to describe the processes of genetic drift and sampling of cases for sequencing (a Kalman filter) (Fig 1A). The hidden states describe the true frequencies of the lineages and the observed states describe the observed frequencies of the lineages as measured via sequenced cases. We adopt Gaussian approximations for the transmission and emission probabilities developed in [75] in order to get analytically tractable forms for the likelihood function, which will greatly speed up our computations.

The transition probability between the true frequencies $f_t$ (the hidden states) due to genetic drift when $\frac{1}{N_e(t)} \ll f \ll 1$ has been shown in [75] to be well-described by the following

expression, which we use as our transition probability,

$$p(f_{t+1}|f_t, \tilde{N}_e(t)) = \frac{1}{2}\sqrt{\frac{2f_t^{1/2}}{\pi f_{t+1}^{3/2}(\tilde{N}_e(t))^{-1}}}\exp\left(-\frac{2(\sqrt{f_{t+1}} - \sqrt{f_t})^2}{(\tilde{N}_e(t))^{-1}}\right). \quad (4)$$

$\tilde{N}_e(t) \equiv N_e(t)\tau(t)$ where $N_e(t)$ is the time-dependent effective population size and $\tau(t)$ is the time-dependent generation time, which is defined as the mean time between two subsequent infections per individual (i.e. the time between when an individual becomes infected and infects another individual, or the time between two subsequent infections caused by the same individual). This transition probability gives the correct first and second moments describing genetic drift when $f \ll 1$, $\mathrm{E}(f_{t+1}|f_t) = f_t$ and $\mathrm{Var}(f_{t+1}|f_t) = \frac{f_t}{N_e(t)}$, and is a good approximation when the central limit theorem can be applied, which is the case when $f \gg 0$. By assuming that $f_{t+1} \approx f_t$, and defining $\phi_t \equiv \sqrt{f_t}$, Eq 4 can be approximated as a simple normal distribution

$$p(\phi_{t+1}|\phi_t, \tilde{N}_e(t)) = \mathcal{N}\left(\phi_t, \frac{1}{4\tilde{N}_e(t)}\right). \quad (5)$$

We describe the emission probability from the true frequency $f_t$ to the observed frequency $f_t^{obs}$ (the observed states), defining $\phi_t^{obs} \equiv \sqrt{f_t^{obs}}$, as

$$p(\phi_t^{obs}|\phi_t, c_t) = \mathcal{N}\left(\phi_t, \frac{c_t}{4M_t}\right) \quad (6)$$

where $M_t$ is the number of input sequences. Again, this distribution is generically a good description when the number of counts is sufficiently large such that the central limit theorem applies (above approximately 20). The first and second moments of this emission probability are $\mathrm{E}(f_t^{obs}|f_t) = f_t$ and $\mathrm{Var}(f_t^{obs}|f_t) = \frac{c_t}{M_t}f_t$, or equivalently considering the number of sequences $n_t^{obs} = f_t^{obs}M_t$ and the true number of positive individuals $n_t$, $\mathrm{E}(n_t^{obs}|n_t) = n_t$ and $\mathrm{Var}(n_t^{obs}|n_t) = c_t n_t$. Thus, $c_t$ describes the strength of measurement noise at time $t$. When $c_t = 1$, the emission probability approaches that describing uniform sampling of sequences from the population of positive individuals (i.e. can be described by a Poisson distribution in the limit of a large number of sequences), namely $\mathrm{Var}(n_t^{obs}|n_t) = n_t$ or equivalently $\mathrm{Var}(f_t^{obs}|f_t) = \frac{f_t}{M_t}$. This is the realistic minimum amount of measurement noise. When $c_t > 1$, it describes a situation where there is bias (that is uncorrelated in time) in the way that sequences are chosen from the positive population. The case of $0 < c_t < 1$ describes underdispersed measurement noise, or noise that is less random than uniform sampling. The case of $c_t = 0$ describes no measurement noise (for instance, when all cases are sampled for sequencing). These last two situations are unlikely in our data, and thus as we describe below, we constrain $c_t \geq 1$ in the inference procedure. In addition to being a good description of measurement noise, defining the emission probability in the same normal distribution form as the transmission probability allows us to easily derive an analytical likelihood function, described below (Note: see Ref. [26] for a method to derive an analytical likelihood function for arbitrary forms of the transition and emission probabilities).

We derive the likelihood function (up to a constant) for the Hidden Markov Model using the forward algorithm, although it can alternatively be derived by marginalizing over all hidden states. We assume an (improper) uniform prior on $\phi_0$ (i.e. no information about the initial

true frequency of the lineage).

$$p(\phi_0, \phi_0^{obs}, \theta_0) = p(\phi_0^{obs}|\phi_0, c_0)p(\phi_0) \tag{7}$$

$$p(\phi_0) \propto 1 \tag{8}$$

$$p(\phi_t, \phi_{0:t}^{obs}, \theta_{0:t}) = p(\phi_t^{obs}|\phi_t, c_t) \int_{-\infty}^{\infty} p(\phi_t|\phi_{t-1}, \tilde{N}_e(t))p(\phi_{t-1}, \phi_{0:t-1}^{obs}, \theta_{0:t-1})d\phi_{t-1}, \quad 0 < t \le T \tag{9}$$

$$p(\phi_{0:T}^{obs}, \theta_{0:T}) = \int_{-\infty}^{\infty} p(\phi_T, \phi_{0:T}^{obs}, \theta_{0:T})d\phi_T \tag{10}$$

$$\mathcal{L}(\vec{\phi}_{0:T}^{obs}|\theta_{0:T}) = \prod_\alpha p(\{\phi_{0:T}^{obs}\}_\alpha, \theta_{0:T})p(\theta_{0:T}) \tag{11}$$

$$p(\theta_{0:T}) \propto 1 \tag{12}$$

$$\mathcal{L}(\vec{\phi}_{0:T}^{obs}|\theta_{0:T}) = \prod_\alpha p(\{\phi_{0:T}^{obs}\}_\alpha, \theta_{0:T}) \tag{13}$$

where $\phi_{0:t}^{obs} \equiv \{\phi_0^{obs}, \ldots, \phi_t^{obs}\}$, $\theta_{0:t} \equiv \{\tilde{N}_e(0), \ldots, \tilde{N}_e(t), c_0, \ldots, c_t\}$, and the subscript $\alpha$ indicates a particular lineage. We use a uniform prior on the parameters. The parameters $\theta_{0:T}$ are inferred by maximizing the likelihood (described below).

The forward algorithm has an analytical form for the simple case of Gaussian transition and emission probabilities. We use the identity for the product of two normal distributions $N(x, \mu, v)$, where $\mu$ is the mean and $v$ is the variance:

$$N(x, \mu_1, v_1)N(x, \mu_2, v_2) = N(\mu_1, \mu_2, v_1 + v_2)N(x, \mu_{12}, v_{12}) \tag{14}$$

$$\mu_{12}(\mu_1, \mu_2, v_1, v_2) = \frac{\mu_1 v_2 + \mu_2 v_1}{v_1 + v_2} \tag{15}$$

$$v_{12}(v_1, v_2) = \frac{1}{\frac{1}{v_1} + \frac{1}{v_2}}. \tag{16}$$

Solving the forward algorithm recursively, we have

$$p(\phi_{0:T}^{obs}, \theta_{0:T}) = \prod_{i=1}^{T} N\left(\phi_i^{obs}, \mu_i, \frac{c_i}{4M_i} + v_i\right) \tag{17}$$

where

$$\mu_1 = \phi_0^{obs} \tag{18}$$

$$\nu_1 = \frac{\frac{1}{N_e(t)} + \frac{c_0}{M_0}}{4} \tag{19}$$

$$\mu_{i+1} = \mu_{12}\left(\mu_i, \phi_i^{obs}, \nu_i, \frac{c_i}{4M_i}\right) \tag{20}$$

$$\nu_{i+1} = \nu_{12}\left(\frac{c_i}{4M_i}, \nu_i\right) + \frac{1}{4\tilde{N}_e(t)}. \tag{21}$$

Eq 17 can be substituted into Eq 13 to obtain the full analytical likelihood function.

## Fitting the model to data

We split the time series data into overlapping periods of 9 Epiweeks, over which the effective population size is assumed to be constant. We first use the moments of the probability distributions combined with least squares minimization to get an initial guess for the parameters. Then, we perform maximum likelihood estimation using the full likelihood function. To capture uncertainties that arise from the formation of coarse-grained lineages from lineages, we create coarse-grained lineages randomly 100 times (except where indicated otherwise). We infer the strength of measurement noise and the effective population size for each coarse-grained lineage combination (described below).

**Determining the initial guess for the parameters using method of moments approach.** Combining the transition and emission probabilities, and marginalizing over the hidden states we have

$$p(f_j^{obs}|f_i^{obs}) \propto \sqrt{\frac{1}{(f_j^{obs})^{3/2}}} \exp\left(-\frac{2\left(\sqrt{f_j^{obs}} - \sqrt{f_i^{obs}}\right)^2}{4\kappa_{i,j}}\right) \tag{22}$$

$$p(\phi_j^{obs}|\phi_i^{obs}) = \mathcal{N}(\phi_i^{obs}, \kappa_{i,j}) \tag{23}$$

$$\kappa_{i,j} \equiv \frac{c_i}{4M_i} + \frac{c_j}{4M_j} + \frac{(j-i)}{4\tilde{N}_e(t)}. \tag{24}$$

The first two terms of $\kappa_{i,j}$ are the contribution to the variance from measurement noise at times $i$ ad $j$, and the third term is the contribution to the variance from genetic drift.

We calculate the maximum likelihood estimate of $\kappa_{i,j}$, $\hat{\kappa}_{i,j}$, which is simply the mean squared displacement

$$\hat{\kappa}_{i,j} = \left\langle (\phi_j^{obs} - \phi_i^{obs})^2 \right\rangle. \tag{25}$$

The standard error is given by

$$\Delta\hat{\kappa}_{i,j} = \sqrt{\frac{\langle[(\phi_j^{obs} - \phi_i^{obs})^2 - \hat{\kappa}_{i,j}]^2\rangle}{Z}} \tag{26}$$

where $Z$ is the number of coarse-grained lineages.

By looking across all pairs of timepoints $i$ and $j$, we get a system of linear equations in $\kappa_{i,j}$ that depend on the parameters $c_t$ and $\tilde{N}_e(t)$. To determine the most likely values of the parameters, we minimize

$$\ln \sum_{i,j} \frac{(\hat{\kappa}_{i,j} - \kappa_{i,j})^2}{\Delta\hat{\kappa}_{i,j}} \tag{27}$$

using scipy.optimize.minimize with the L-BFGS-B method and the bounds $1 \leq c_t \leq 100$ and $1 \leq \tilde{N}_e(t) \leq 10^7$. While underdispersed measurement noise ($c_t < 1$) is in principle possible, we constrain $c_t \geq 1$ because realistically, the lowest amount of measurement noise will be from uniform sampling of sequences. An example of inferred parameters using the methods of moments approach on simulated data is shown in S23 Fig.

**Maximum likelihood estimation of the parameters.** For each set of coarse-grained lineages, we use the inferred measurement noise values ($c_t$) and inferred scaled effective population size from above ($\tilde{N}_e(t)$) as initial guesses in the maximization of the likelihood function in Eq 13 over the parameters. For the optimization, we use scipy.optimize.minimize_scalar with the Bounded method and the bounds $1 \leq c_t \leq 100$ and $1 \leq \tilde{N}_e(t) \leq 10^{11}$. The time $t$ in the inferred $\tilde{N}_e(t)$ is taken to be the midpoint of the 9 Epiweek period. The reported $\tilde{N}_e(t)$ is the median inferred $\tilde{N}_e(t)$ across all coarse-grained lineage combinations where $\tilde{N}_e(t) < 10^5$ (values above $10^5$ likely indicate non-convergence of the optimization, because most values above $10^5$ are at $10^{11}$, see S24 Fig). The reported errors on $\tilde{N}_e(t)$ are the 95% confidence intervals (again taking the median across all coarse-grained lineage combinations where $\tilde{N}_e(t) < 10^5$) which are calculated by using the likelihood ratio to get a p-value [76, 77]. We replace the likelihood with the profile likelihood, which has the nuisance parameters $c_{0:T}$ profiled out:

$$p > 0.05 \tag{28}$$

$$p = \int I\left[\frac{\mathcal{L}_{\tilde{N}_e}(\hat{c}_{0:T}|\vec{\phi}_{0:T}^{obs})}{\mathcal{L}_{\tilde{N}_e'}(\hat{c}_{0:T}|\vec{\phi}_{0:T}^{obs})} > 1\right] P_{\tilde{N}_e'}(\hat{c}_{0:T}|\vec{\phi}_{0:T}^{obs})d\tilde{N}_e' \tag{29}$$

$$\hat{c}_{0:T} = \arg\max_{c_{0:T}} \mathcal{L}_{\tilde{N}_e}(c_{0:T}|\vec{\phi}_{0:T}^{obs}) \tag{30}$$

$$P_{\tilde{N}_e'}(\hat{c}_{0:T}|\vec{\phi}_{0:T}^{obs}) \propto \mathcal{L}_{\tilde{N}_e'}(\hat{c}_{0:T}|\vec{\phi}_{0:T}^{obs})p(\tilde{N}_e) \tag{31}$$

$$p(\tilde{N}_e) \propto 1 \tag{32}$$

where $I$ is an indicator function that equals one when the argument is true and zero otherwise, $\mathcal{L}_{\tilde{N}_e}(\hat{c}_{0:T}|\vec{\phi}_{0:T}^{obs})$ is the profile likelihood with the nuisance parameters (in this case) $c_{0:T}$ profiled out, $P_{\tilde{N}_e'}(\hat{c}_{0:T}|\vec{\phi}_{0:T}^{obs})$ is the posterior where we have used a uniform prior. We also tried a Jeffreys prior which is used for variance parameters, but it gave similar results on simulated data because it looked relatively flat over the values of $\tilde{N}_e(t)$ of interest. As the Jeffreys prior was

more computationally expensive than the uniform prior and the two priors gave similar results, we used the uniform prior for the analyses.

The reported values of $c_t$ are the median across all coarse-grained lineage combinations and across all time series segments where the timepoint appears. The reported errors on $c_t$ are the 95% confidence intervals as calculated by the middle 95% of values across coarse-grained lineage combinations and time series segments.

We checked that if we allow $c_t \geq 0$, the results are similar to if we constrain $c_t \geq 1$ (compare Fig 2 and S25 Fig).

An example of inferred parameters on simulated data using the maximum likelihood estimation approach, compared to the initial guesses of the parameters from the methods of moments approach, is shown in S23 Fig.

## Correcting for the number of sequences assigned to lineages

Because some sequences occur before the cut point in the tree that is used for creating lineages, they are not included in any lineages. As a result, the number of sequences assigned to lineages is lower than the number of sequences in the tree (S26 Fig). This will bias the inferred $\tilde{N}_e(t)$ to be lower than in reality when the omitted sequences are from a particular part of the tree even when the dynamics are neutral (i.e. a certain part of the population is being left out of the analysis). To correct for the bias in inferred effective population size that results from leaving out sequences from parts of the tree, we divide the inferred effective population size by the fraction of sequences in the tree that are assigned to a lineage. We note that while the number of sequences in the tree is less than the total number of sampled sequences, the sequences in the tree were chosen to be a representative fraction of the total sampled sequences. Thus, we do not need to additionally correct for the downsampling of sequences that were included in the tree. To test that randomly subsampling sequences for the analysis does not affect the results, we randomly subsampled half of the Delta sequences, and reran the analyses; the inferred effective population size was very similar to that from the full number of sequences (S27 Fig).

## Simulations for validating method

For the model validation, we perform simulations of the lineage trajectories using a discrete Wright-Fisher model. 500 lineages are seeded initially, and the initial frequency of lineages is taken to be the same across all lineages. In each subsequent Epiweek, the true number of counts for a lineage is drawn from a multinomial distribution where the probabilities of different outcomes are the true frequencies of the lineages in the previous Epiweek and the number of experiments is the effective population size. The true frequency is calculated by dividing the true number of counts by $N$. The observed counts are drawn from a negative binomial distribution,

$$p(n_t^{obs}|f_t) = NB(r, q) \equiv \binom{n_t^{obs} + r - 1}{r - 1} q^r (1 - q)^{n_t^{obs}} \tag{33}$$

$$r = \frac{f_t M_t}{c_t - 1} \tag{34}$$

$$q = \frac{1}{c_t} \tag{35}$$

which has the same mean and variance as the emission probability in Eq 6. The total number

of observed sequences in each timepoint is calculated empirically after the simulation is completed, as it may not be exactly $M_t$. The simulation is run for 10 weeks of "burn-in" time before recording to allow for equilibration. Coarse-grained lineages are created in the same way as described above.

For long time series simulations, some lineages will go extinct due to genetic drift, making it challenging to have sufficient data for the analysis. To be able to have a high enough number of lineages for the entire time series, we introduce mutations that lead to the formation of a new lineage with a small rate $\mu = 0.01$ per generation per individual.

## Simulations for testing the effect of balancing selection

For the simulations that test for the effect of balancing selection, the simulations described above were modified as follows. Initially, each individual has a fitness drawn from the empirical distribution of deleterious fitness effects. Additionally, each individual forms a single lineage. To model selection, the probability of being drawn in the multinomial distribution is weighted by $e^s$, where $s$ is the fitness coefficient. Mutations occur on the background of each individual in each generation with probability 0.01 and the mutants have a fitness that is the sum of that of the parent and a newly drawn fitness from the distribution of deleterious fitness effects. The burn-in period ends when the number of lineages reaches the threshold of 100 lineages, and recording begins. No new lineages are created in the simulation, so lineages are defined as the descendants of the individuals that are initially in the simulation.

## Calculating the effective population size for an SIR or SEIR model

The effective population size times the generation time in an SIR model is given by Refs. [37, 43]

$$\tilde{N}_e^{\text{SIR}}(t) \equiv N_e^{\text{SIR}}(t)\tau(t) = \frac{I(t)}{2R_t\gamma_I}. \tag{36}$$

The variance in offspring number for an SIR model is approximately 2.

For an SEIR model, we calculated $\tilde{N}_e(t)$ following the framework from Ref. [38]. Using this framework, we were only able to consider a situation where the epidemic is in equilibrium. We test how well this approximates the situation out of equilibrium using simulations (see S1 Appendix).

We first considered how the mean number of lineages, $A$, changes going backwards in time, $s$, which is given by

$$\frac{dA}{ds} = -fp_c \tag{37}$$

where $f$ is the number of transmissions per unit time and $p_c$ is the probability that a transmission results in a coalescence being observed in our sample. $p_c$ is given by the number of ways of choosing two lineages divided by the number of ways of choosing two infectious individuals

$$p_c = \frac{\binom{A(s)}{2}}{\binom{N(s)}{2}} \stackrel{\lim_{N(s)\to\infty}}{=} \binom{A(s)}{2}\frac{2}{N(s)^2}. \tag{38}$$

where the limit assumes that the number of infectious individuals, $N(s)$, is large. In the

Kingman coalescent we also have

$$\frac{dA}{ds} = -\binom{A(s)}{2} \frac{1}{\tilde{N}_e(t)}.$$ (39)

Combining Eqs 37, 38 and 39, we have

$$\tilde{N}_e(t) = \frac{N(s)^2}{2f}.$$ (40)

Thus by determining the number of transmissions per unit time, $f$, and the number of infectious individuals, $N(s)$, in an SEIR model, we can find an expression for $\tilde{N}_e(t)$.

These quantities can be derived from the equations describing the number of susceptible ($S$), exposed ($E$), infectious ($I$), and recovered ($R$) individuals in an SEIR model

$$\frac{dS}{dt} = -\beta I \frac{S}{N_H}$$ (41)

$$\frac{dE}{dt} = \frac{\beta I S}{N_H} - \gamma_E E - \delta_E E$$ (42)

$$\frac{dI}{dt} = \gamma_E E - \gamma_I I - \delta_I I$$ (43)

$$\frac{dR}{dt} = \gamma_I I$$ (44)

where $\beta$ is the number of transmissions per infectious individual per unit time (the number of contacts made by an infectious individual per unit time multiplied by the probability that a contact results in a transmission), $N_H$ is the total population size ($N_H = S + E + I + R$), $\gamma_E$ is the rate that an exposed individual becomes infectious, $\delta_E$ is the rate of death for an exposed individual, $\gamma_I$ is the rate than an infectious individual recovers, and $\delta_I$ is the rate of death for an infectious individual.

The number of infectious individuals in a generation, $N(s)$, is given by the instantaneous number of infectious individuals plus the number of exposed individuals that will become infectious in that generation [43]. Thus,

$$N(s) = \frac{\gamma_E}{\gamma_E + \delta_E} E + I.$$ (45)

The number of transmissions per unit time is given by

$$f = \beta I \frac{S}{N_H}.$$ (46)

We rewrite $f$ in terms of the effective reproduction number (for which data are available) which is given by the number of transmissions per unit time ($f$) divided by the number of recoveries and deaths per unit time

$$R_t = \frac{f}{(\gamma_I + \delta_I)I + \delta_E E}.$$ (47)

Putting everything together, we have that $\tilde{N}_e(t)$ for an SEIR model in equilibrium is given by

$$\tilde{N}_e^{\text{SEIR,eq}}(t) = \frac{\left[\left(\frac{\gamma_E}{\gamma_E+\gamma_I}\right)E + I\right]^2}{2R_t[(\gamma_I + \delta_I)I + \delta_E E]}. \tag{48}$$

For SARS-CoV-2, the death rates are much lower than the rate at which exposed individuals become infectious and the rate at which infectious individuals recover ($\delta_E, \delta_I \ll \gamma_E, \gamma_I$). In this limit, Eq 48 simplifies to

$$\tilde{N}_e^{\text{SEIR,eq}}(t) = \frac{(E+I)^2}{2R_t\gamma_I I}. \tag{49}$$

To calculate the $\tilde{N}_e$ for an SIR or SEIR model, we use the estimated number of positives from the COVID-19 Infection Survey for $I(t)$. This number is an estimate of the number of positive individuals in the community as measured by surveillance and includes both symptomatic and asymptomatic individuals. While the estimated number of positives does not include cases from hospitals, care homes, and other communal establishments, community cases likely contribute the most to transmission. We used the measured effective reproduction number from the UK Health Security Agency for $R_t$.

To calculate the number of exposed individuals for the SEIR model, we solved for $E$ in Eq 43 (taking $\delta_E \ll \gamma_E$)

$$E = \frac{1}{\gamma_E}\left(\frac{dI}{dt} + \gamma_I I\right). \tag{50}$$

$\frac{dI}{dt}$ was calculated numerically as $\frac{I(t+\Delta t) - I(t-\Delta t)}{2\Delta t}$ where $\Delta t = 1$ week. The parameter values used were $\gamma_E^{-1} = 3$ days and $\gamma_I^{-1} = 5.5$ days [41, 42]. We checked that varying the value used for $\gamma_I$ does not substantially affect the results (S11 Fig). The error on $E$ was calculated by taking the minimum and maximum possible values from the combined error intervals of $I(t + \Delta t)$ and $I(t - \Delta t)$ (note that this does not correspond to a specific confidence interval size).

The error on $\tilde{N}_e(t)$ for the SIR or SEIR model was calculated similarly by taking the minimum and maximum possible values from the combined error intervals of $E$, $I$, and $R_t$. Only time points where the error interval of $\tilde{N}_e(t)$ was less than 3 times the point estimate were kept.

## Calculating the effective population size for an SIR or SEIR model by variant

To calculate the effective population size for an SIR or SEIR model by variant, we needed to determine the variant-specific: number of infectious individuals $I(t)$, number of exposed individuals $E(t)$, effective reproduction number $R_t$, and rate than an infectious individual recovers $\gamma_I$. We assumed that $\gamma_I$ is constant between variants. We calculated the number of infectious individuals $I(t)$ by multiplying the total number of positives by the fraction of each variant in the reported sequences. This should be a good representation of the fraction of the variant in the population as the sequences are a random sample of cases detected via surveillance. We calculated the number of variant-specific exposed individuals $E(t)$ in the same way as described above using the variant-specific number of infectious individuals. We assumed that the rate an exposed individual becomes infectious $\gamma_E$ is constant between variants.

We calculated the variant-specific effective reproduction number by rescaling the measured effective reproduction number for the whole population

$$R_t^v = R_t \frac{R_0^v}{\sum_w R_0^w f^w} \tag{51}$$

where $R_0^w$ is the basic reproduction number of the variant $w$ and $f^w$ is the fraction of the infectious population with variant $w$. The values of $R_0$ when rescaled to $R_0^{pre-B.1.177}$ that are used for the data presented in the main text are $\frac{R_0^{pre-B.1.177}}{R_0^{pre-B.1.177}} = \frac{R_0^{B.1.117}}{R_0^{pre-B.1.177}} = 1$, $\frac{R_0^{Alpha}}{R_0^{pre-B.1.177}} = 1.7$ (Ref. [17]), $\frac{R_0^{Delta}}{R_0^{pre-B.1.177}} = 1.97$ (Ref. [78]). We assumed the same $R_0$ for pre-B.1.177 and B.1.177 since the B.1.177 variant was shown to have increased in frequency due to importations from travel rather than increased transmissibility [45]. Varying the variant $R_0$ within the ranges reported in the literature does not substantially affect the results (S28 Fig).

## Inference of fitness from lineage frequency time series

We sought to infer the fitness effects of individual lineages, so that we could then determine if putatively selected lineages are influencing the estimation of the time-varying effective population sizes. We used a deterministic method to estimate lineage fitness effects, similar to the method described in [79].

On average, when the frequency of lineage $i$ is sufficiently small $f_{t,i} \ll 1$, the frequency dynamics will exponentially grow/decay according to the lineage fitness effect, $s_i$,

$$\langle f_{t,i} \rangle = f_{0,i} e^{s_i t}$$

The two sources of noise–genetic drift and measurement noise–both arise from counting processes, so the combined noise will follow var $(f_{t,i}) \propto \langle f_{t,i} \rangle$. To account for the inherent discreteness of the number of cases in a lineage–especially important to accurately model lineages at low frequencies–we modeled the observed counts at Epiweek $t$ of lineage $i$, $r_{t,i}$, as a negative binomial random variable,

$$r_{t,i} | s_i, f_{0,i} \sim \mathrm{NB}\left(\mu_{t,i}, \zeta_t\right) \tag{52}$$

$$\langle r_{t,i} \rangle = \mu_{t,i} \tag{53}$$

$$\mathrm{var}\left(r_{t,i}\right) = \zeta_t \langle r_{t,i} \rangle \tag{54}$$

$$\mu_{t,i} = M_t f_{0,i} e^{s_i t} \tag{55}$$

Where $M_t$ is the total number of sequences, and $\zeta_t$ is a dispersion parameter. We took $\zeta_t$ as the total marginal variance at a given time-point, i.e. $\zeta_t = c_t + M_t/N_e(t)$, where we computed estimates of $c_t$ and $N_e$ as previously described (section "Maximum likelihood estimation of the parameters"). The final likelihood for the fitness, $s_i$, of lineage $i$ is obtained by combining the data from all the relevant the time-points,

$$P(\boldsymbol{r}_i | s_i, f_{0,i}) = \prod_t \frac{\Gamma\left(r_{t,i} + \frac{\mu_{t,i}}{\zeta_t - 1}\right)}{\Gamma\left(\frac{\mu_{t,i}}{\zeta_t - 1}\right)\Gamma\left(r_{t,i} + 1\right)} \frac{(\zeta_t - 1)^{r_{t,i}}}{\zeta_t^{r_{t,i} + \frac{\mu_{t,i}}{\zeta_t - 1}}} \tag{56}$$

The point estimate of the lineage fitness, $\hat{s}_i$, is then numerically computed as the maximum likelihood,

$$\hat{s}_i = \operatorname*{argmax}_{s_i} \log P(\boldsymbol{r}_i | s_i, f_{0,i}). \tag{57}$$

## Stochastic simulations of SEIR model

The stochastic simulations of an SEIR model were performed using a Gillespie simulation with 4 states: susceptible, exposed, infectious, and recovered, where the number of individuals in each state are denoted by $S(t)$, $E(t)$, $I(t)$, and $R(t)$ respectively. There are 3 types of events that lead to the following changes in the number of individuals in each state

1. Infection of an susceptible individual with probability $\frac{\beta I(t) S(t)}{N(t)}$

$$S(t) = S(t) - 1 \tag{58}$$

$$E(t) = E(t) + 1 \tag{59}$$

2. Transition of an exposed individual to being infectious with probability $\gamma_E E(t)$

$$E(t) = E(t) - 1 \tag{60}$$

$$I(t) = I(t) + 1 \tag{61}$$

3. Recovery of an infectious individual with probability $\gamma_I I(t)$

$$I(t) = I(t) - 1 \tag{62}$$

$$R(t) = R(t) + 1 \tag{63}$$

where $\beta \equiv R_0 \gamma_I$, $R_0$ is the basic reproduction number, $\gamma_E$ is the rate that exposed individuals become infectious, and $\gamma_I$ is the rate that infectious individuals recover. As in the rest of this work, we assume that the birth rate of susceptible individuals, background death rate, and the death rate due to disease are much slower compared to the rates of the above processes and thus can be neglected from the dynamics.

The time until the next event is drawn from an exponential distribution with rate given by the inverse of the sum of the above probabilities, and the type of event is randomly drawn weighted by the respective probabilities.

Because the time of the events occurs in continuous time, but the inference method of the effective population size works in discrete time, we must convert from continuous to discrete time. To perform this conversion, we calculate the net number of events of each type in each chosen unit of discrete time (1 week) and perform the changes in the number of individuals of each state as described above. Thus, for example, if within the same week an individual becomes exposed and then becomes infectious, it will cause the number of susceptible individuals to decrease by 1, no change in the number of exposed individuals, and the number of infectious individuals to increase by 1.

The infected (or infected and exposed) individuals are randomly assigned a lineage at a given time after the start of the epidemic. For our simulations, we chose the lineage labeling time as 75 days or 10.7 weeks since the approximate number of infectious individuals was high enough at that time to generate sufficient diversity in lineages, and we chose the number of different types of lineages as 100. The other parameters that we used for the simulations were $R_0 = 2$, $\gamma_E^{-1} = 3$ days, $\gamma_I^{-1} = 5.5$ days, $N(t) = S(t) + E(t) + I(t) + R(t) = 10^6$. The initial condition of the simulation is $S(t) = N(t) - 1$, $E(t) = 1$, and $I(t) = R(t) = 0$.

To test the sensitivity of the results to whether the reported PCR positive individuals are infectious or whether they can also be from the exposed class, we recorded the results in two ways. In the first case, only the infectious individuals we recorded as positive (S29 Fig), and in the second case both the exposed and infectious individuals were recorded as positive (S30 Fig). Inference of $\tilde{N}_e(t)$ was subsequently done on the lineage frequency trajectories of the recorded positive individuals. The SIR or SEIR model $\tilde{N}_e(t)$ were calculated analytically using the true numbers of infectious and exposed individuals and numerically using the number of positive individuals as described above in "Calculating the effective population size for an SIR or SEIR model".

## Deme simulations

To better understand the effect of host population structure on the effective population size, we simulated a simple situation where there are "demes", or groups, of individuals with very high rates of transmission between individuals in that deme, but the rate of transmission between individuals from different demes is very low. In a given simulation, all demes have the same number of individuals (10, 50, 100, or 200). The total number of demes is chosen to be very high ($5.6 \times 10^6$). Initially, a certain number of demes (100, 1000, 2000, or 5000) are each seeded by a single infectious individual infected by a randomly chosen lineage (200 different lineages). We simulated deterministic SEIR dynamics within demes with $R_0 = 10$, $\gamma_E = (2.5$ days$)^{-1}$, $\gamma_I = (6.5$ days$)^{-1}$. We simulated Poisson transmission dynamics between demes. In order to calibrate the overall population dynamics to be roughly in equilibrium (the number of infectious individuals is not deterministically growing or shrinking), we draw the number of between-deme infections caused by a given deme from a Poisson distribution with mean 1. The time of the between-deme infection event is randomly chosen, weighted by the number of infected individuals within a deme at a given time. The number of infectious individuals in each lineage is recorded every 1 week, and the frequency of the lineage is calculated by dividing by the total number of infectious individuals from all lineages in that week. The lineage frequency data from a period of 9 weeks starting in week 42 is used for the inference of effective population size. In this time period, only a small number of demes have been infected such that the total number of demes did not matter. The effective population size inference is performed as above except in the absence of measurement noise, so there is no emission step in the HMM.

## Supporting information

**S1 Acknowledgments. Membership of the COVID-19 Genomics UK (COG-UK) Consortium.**
(DOCX)

**S1 Appendix. Supplementary information.**
(PDF)

**S1 Table. Overdispersion values from the literature ordered from highest to lowest variance in offspring number.** Any error intervals that are reported are taken from the reference (sometimes defined differently). The estimate taken from Ref. [57] assumes no self-isolation upon symptom onset and no testing; lifting these assumptions leads to similar or lower overdispersion.
(PDF)

**S1 Fig. The fraction of simulations (20 total) where the inferred 95% confidence interval for $\tilde{N}_e(t)$ or $c_t$ included the true value (left) by timepoint and (right) for all timepoints.** (Right) Boxes indicate the quartiles and the line inside the box (and number above) indicates the median. Whiskers indicate the extreme values excluding outliers. Simulation parameters are specified in the Methods and Fig 1, which shows a single simulation instance. For the inference, we created coarse-grained lineages randomly 20 times.
(PDF)

**S2 Fig. Wright-Fisher simulations where $\tilde{N}_e(t)$ is constant over time, and the inferred $\tilde{N}_e(t)$ and $c_t$.** (a) Number of sequences sampled. (b) Simulated lineage frequency trajectories. (c) Inferred effective population size ($\tilde{N}_e(t)$) on simulated data compared to true values. (d) Inferred measurement noise ($c_t$) on simulated data compared to true values. In (c) the shaded region shows the 95% confidence interval calculated using the posterior, and in (d) the shaded region shows the 95% confidence interval calculated using bootstrapping (see Methods).
(PDF)

**S3 Fig. Wright-Fisher simulations where $\tilde{N}_e(t)$ changes over time according to a rectangular function, and the inferred $\tilde{N}_e(t)$ and $c_t$.** (a) Number of sequences sampled. (b) Simulated lineage frequency trajectories. (c) Inferred effective population size ($\tilde{N}_e(t)$) on simulated data compared to true values. (d) Inferred measurement noise ($c_t$) on simulated data compared to true values. In (c) the shaded region shows the 95% confidence interval calculated using the posterior, and in (d) the shaded region shows the 95% confidence interval calculated using bootstrapping (see Methods).
(PDF)

**S4 Fig. Total number of surveillance sequences of each variant in the metadata from COG-UK downloaded on January 16, 2022 and the number of sequences used in the analysis for each variant or group of lineages (determined by the number of sequences included in the tree, and the number of sequences which could be grouped into lineages based on the procedure described in the Methods).**
(PDF)

**S5 Fig. Varying the date of the tree downloaded from COG-UK and the depth at which the tree is cut for creating lineages ($d_{\text{cut}}$, which is defined as the number of mutations from the root of the tree, see Methods) does not substantially change the inferred scaled effective population size.** The tree date and depth used in the main text are {2021-02-22, B.1.177, $d_{\text{cut}} = 2.323 \cdot 10^{-2}$}, {2021-06-20, Alpha, $d_{\text{cut}} = 2.054 \cdot 10^{-3}$}, {2022-01-25, Delta, $d_{\text{cut}}^{(1)} = 1.687 \cdot 10^{-3}$, $d_{\text{cut}}^{(2)} = 1.954 \cdot 10^{-3}$}. The color of the lines for the parameters that were used in the main text are the same as those shown in Fig 2.
(PDF)

**S6 Fig. The inferred effective population size when cutting the tree at different depths to test the effect of combining lineages with other more closely related lineages in forming**

the coarse-grained lineages.
(PDF)

**S7 Fig. The lineage frequency time series using the tree cut depths shown in S6 Fig.**
(PDF)

**S8 Fig. Varying the threshold counts for forming coarse-grained lineages (see Methods) does not substantially change the inferred scaled effective population size.** The coarse-grained lineage threshold counts used in the main text is 20.
(PDF)

**S9 Fig. Varying the number of weeks in the moving window does not substantially change the inferred scaled effective population size.** The size of the moving window used in the main text is 9 weeks.
(PDF)

**S10 Fig. The distribution of square root observed frequency displacements** $(\sqrt{f_{t+1}^{obs}} - \sqrt{f_t^{obs}})$ **across all time points normalized by the inferred variance due to genetic drift and measurement noise** $(\kappa_{t,t+1} = \frac{c_t}{4M_t} + \frac{c_{t+1}}{4M_{t+1}} + \frac{1}{N_e(t)}$, **see Eq 24).** The orange line is a plot of a normal distribution with mean 0 and variance 1.
(PDF)

**S11 Fig. Varying the rate of transitioning from infected to recovered within literature ranges ($\gamma_I$ = 3 to 14 days) used for calculation of the SIR model $\tilde{N}_e(t)$ (Methods) does not substantially decrease the observed ratio $\tilde{N}_e^{\text{SIR}}(t)/\tilde{N}_e^{\text{inf}}(t)$.**
(PDF)

**S12 Fig. Inferred scaled effective population size compared to the SIR model scaled population size calculated using the observed number of positive individuals in England (see Methods).**
(PDF)

**S13 Fig. Inferred scaled effective population size compared to the SEIR model scaled population size calculated using the observed number of positive individuals in England (see Methods).**
(PDF)

**S14 Fig. The inferred effective population size when excluding beneficial lineages whose inferred absolute fitness value are above the 50th ($|s| > 0.09$), 75th ($|s| > 0.16$), and 90th ($|s| > 0.27$) percentiles compared to that when all lineages are included.**
(PDF)

**S15 Fig. The distribution of deleterious fitness effects from Ref. [44].** The orange vertical line indicates $\frac{1}{N}$, which is the threshold in fitness above which selection dominates over genetic drift. Here, $N$ is set to $10^4$, which is the order of magnitude of the census population size of SARS-CoV-2 in England.
(PDF)

**S16 Fig. Simulated lineage frequency dynamics where deleterious mutations occur at rate 0.01/genome/generation and the distribution of deleterious fitness effects is taken from the empirically estimated values in Ref. [44].** The inferred effective population size and measurement noise are shown.
(PDF)

**S17 Fig. The same simulation as in S16 Fig but as a control, where the fitness of new mutations is always 0.** The inferred effective population size and measurement noise are shown. (PDF)

**S18 Fig. The cumulative mean ratio of the point estimates of the inferred effective population size in the simulations using the empirical distribution of deleterious fitness effects and the neutral simulations.**
(PDF)

**S19 Fig. Inferred effective population size in regions of England.** (Top panels) Inferred $\tilde{N}_e(t)$ of pre-B.1.177 lineages, B.1.177, Alpha, and Delta for each region of England. The inferred $\tilde{N}_e(t)$ for England as a whole is shown for reference. Shaded regions show 95% confidence intervals (see Methods). (Bottom panels) The ratio between the inferred $\tilde{N}_e(t)$ of England and that of the region for each variant. A horizontal dashed line indicates a ratio of 1 (i.e. $\tilde{N}_e(t)$ is the same in that region of England and England as a whole). Shared regions show the minimum and maximum possible values of the ratio from the combined error intervals of the numerator and denominator (thus, not corresponding to a specific confidence interval range).
(PDF)

**S20 Fig. Inferred scaled effective population size by region in England, compared to number of positives at the community level in that region reported by the COVID-19 Infection Survey [36].**
(PDF)

**S21 Fig. Simulations of deme structure (described in main text and Methods).** (a) The mean number of infected individuals per week from Weeks 42 to 50. (b) The inferred $\tilde{N}_e(t)$ using lineage trajectories from Weeks 42 to 50.
(PDF)

**S22 Fig. Sample epiweeks versus tree depths.** In a phylogenetic tree, the number of sequences (leaf nodes) of a focal variant that fall within specific epiweek and tree depth ranges is counted and summarized as a two-dimensional histogram. The tree depth is the substitution rate measured in units of substitutions per site, with respect to the most recent common ancestor. From left to right, the phylogenetic tree (specified by date created by COG-UK, using the sequences available at the time) and focal variant are {2021-02-22, B-1-177}, {2021-06-01, Alpha}, {2021-06-20, Alpha}, and {2022-01-25, Delta}. Weeks are counted from 2019-12-29. The dashed horizontal lines indicate the values of $d_{\text{cut}}$ ($d_{\text{cut}}^{(1)}$ and $d_{\text{cut}}^{(2)}$ for the Delta variant) used for the results presented in the main text, except for the 2021–06-01 Alpha tree, where they indicate the value of $d_{\text{cut}}$ tested in S5 Fig.
(PNG)

**S23 Fig. Comparing the inferred $\tilde{N}_e(t)$ and $c_t$ in Wright-Fisher simulations using the method of moments and maximum likelihood estimation approaches (see Methods).** (a) Number of sequences sampled. (b) Simulated lineage frequency trajectories. (c) Inferred effective population size ($\tilde{N}_e(t)$) on simulated data using the method of moments (MSD, for mean squared displacement) and maximum likelihood (HMM, for Hidden Markov Model) estimation approaches compared to true values. The shaded region shows the 95% confidence interval of the inferred values. The confidence interval using the method of moments approach was calculated by taking the middle 95% of values when bootstrapping over the coarse-grained lineages. The confidence interval using the maximum likelihood estimation approach was

determined using the posterior (see Methods) and takes into account joint errors in $c_t$ and $\tilde{N}_e(t)$. (d) Inferred measurement noise ($c_t$) on simulated data using the method of moments and maximum likelihood estimation approaches compared to true values. The shaded region shows the 95% confidence interval calculated using bootstrapping (see Methods).
(PDF)

**S24 Fig. Inferred effective population size from different times and coarse-grained lineage combinations.** The vertical dashed line indicates $10^5$ which is the value above which results in the text were thrown away due to non-convergence (these only include values at $10^{11}$).
(PDF)

**S25 Fig. The inferred measurement noise overdispersion parameter for England as a whole when changing the lower bound of the overdispersion parameter from 1 to 0.**
(PDF)

**S26 Fig. The fraction of sequences in the tree that are assigned to a lineage.** The blue shading indicates the period of time in the data that was used for the inference analysis.
(PDF)

**S27 Fig. Randomly subsampling half of the Delta sequences used for the analysis does not substantially change the inferred scaled effective population size.**
(PDF)

**S28 Fig. Varying the values of the basic reproduction number within literature ranges ($\frac{R_0^{\text{Alpha}}}{R_0^{\text{pre-B.1.1.7}}} = 1.1 - 2.7$ [17], $\frac{R_0^{\text{Delta}}}{R_0^{\text{pre-B.1.1.7}}} = 1.76 - 2.17$ [78]) used for calculation of the SIR model $\tilde{N}_e(t)$ by variant (Methods) does not substantially affect the calculated $\tilde{N}^{\text{SIR}}(t)$.**
(PDF)

**S29 Fig. Simulations of stochastic SEIR dynamics without measurement noise, and comparison of the inferred $\tilde{N}_e(t)$ to Eqs 1 and 49 when the reported positive individuals include only the infectious individuals.** (Top) Muller plot of simulated infectious individuals' lineage trajectories (simulations described in Methods). Infectious individuals are randomly assigned a lineage in week 11, and individuals that they transmit to are infected with the same lineage. The blue lineage before week 11 indicates the infectious individuals that existed before lineages were assigned. (Bottom) Comparison of the inferred $\tilde{N}_e(t)$ using the lineage trajectories shown in the top panel to the number of infectious individuals $I(t)$, Eq 49 (SEIR model $\tilde{N}_e(t)$ at equilibrium), and Eq 1 (SIR model $\tilde{N}_e(t)$) calculated analytically or numerically as described in the Methods. The numerical solutions give the same results as the analytical solutions.
(PDF)

**S30 Fig. Simulations of stochastic SEIR dynamics without measurement noise, and comparison of the inferred $\tilde{N}_e(t)$ to Eqs 1 and 49 when the reported positive individuals include both infectious and exposed individuals.** (Top) Muller plot of simulated infectious and exposed individuals' lineage trajectories (simulations described in Methods). Infectious and exposed individuals are randomly assigned a lineage in week 11, and individuals that they transmit to are infected with the same lineage. The blue lineage before week 11 indicates the infectious and exposed individuals that existed before lineages were assigned. (Bottom) Comparison of the inferred $\tilde{N}_e(t)$ using the lineage trajectories shown in the top panel to the number of infectious individuals $I(t)$, the sum of the number of infectious and exposed individuals $I(t) + E(t)$, Eq 49 (SEIR model $\tilde{N}_e(t)$), and Eq 1 (SIR model $\tilde{N}_e(t)$) calculated analytically or

numerically as described in the Methods. The numerical solutions give slightly higher $\tilde{N}_e(t)$ as compared with the analytical solutions; however, the numerical solutions to the SEIR and SIR models bound the inferred $\tilde{N}_e(t)$.
(PDF)

**S31 Fig. The effect of the empirically estimated distribution of deleterious fitness effects in SARS-CoV-2 [44] on the effective population size using the analytical theory derived in Ref. [80] (Equation 4 in S1 Appendix).** In this calculation, the effective population size in the absence of background selection is $10^4$, the clock rate is 31 substitutions per year, and the generation time is 5.1 days.
(PDF)

**S32 Fig. Inferred scaled effective population size by region in England, compared to that of an SIR model as calculated using the observed number of positives at the community level in that region reported by the COVID-19 Infection Survey [36] and the observed effective reproduction number in that region reported by the UK Health Security Agency [40].**
(PDF)

**S33 Fig. Inferred measurement noise by region in England.**
(PDF)

**S34 Fig. Same as Fig 3c, but plotting the overdispersion parameter, $k = \frac{R_t}{\frac{\sigma^2}{R_t}-1}$, where $R_t$ is the effective reproduction number and $\sigma^2$ is the variance in offspring number.** The circles show the inferred overdispersion parameter if we assume there is only superspreading and no deme structure. For the inferred overdispersion parameter, the estimated effective reproduction number in England by variant (see Methods) is used for $R_t$, and the ratio between the SIR model population size and the inferred effective population size is used for $\sigma^2$. The shaded area for the inferred overdispersion parameter $k$ gives an estimate of the error and is calculated by combining minimum or maximum values of the individual parameters; note that this does not correspond to a particular confidence interval.
(PDF)

## Acknowledgments

We are grateful to the Hallatschek lab for helpful discussions, feedback, and comments on earlier versions of this manuscript, particularly Giulio Issachini and Valentin Slepukhin. We are grateful to Aditya Prasad for advice on computing. We thank Mike Boots, Vince Buffalo, Katia Koelle, Priya Moorjani, Rasmus Nielsen, Daniel Reeves, and Daniel Weissman for helpful discussions and feedback. We thank Hernan G. Garcia and Yun S. Song for helpful comments on earlier versions of this manuscript. This research used resources of the National Energy Research Scientific Computing Center (NERSC, https://www.nersc.gov/), a U.S. Department of Energy Office of Science User Facility located at Lawrence Berkeley National Laboratory, operated under Contract No. DE-AC02-05CH11231 using NERSC BER-ERCAP0019907. The authors acknowledge use of data generated through the COVID-19 Genomics Programme funded by the Department of Health and Social Care (https://www.gov.uk/government/organisations/department-of-health-and-social-care). We thank the COG-UK consortium and all partners and contributors who are listed at https://webarchive.nationalarchives.gov.uk/ukgwa/20230507113711/https://www.cogconsortium.uk/about/about-us/about-us/ and in S1 Acknowledgments.

## Disclaimer

The views expressed are those of the author and not necessarily those of the Department of Health and Social Care or UK Health Security Agency.

## Author Contributions

**Conceptualization:** QinQin Yu, Joao A. Ascensao, Takashi Okada, Oskar Hallatschek.

**Data curation:** QinQin Yu, Takashi Okada, Olivia Boyd.

**Formal analysis:** QinQin Yu, Joao A. Ascensao, Takashi Okada, Oskar Hallatschek.

**Funding acquisition:** Oskar Hallatschek.

**Investigation:** QinQin Yu, Joao A. Ascensao, Takashi Okada, Oskar Hallatschek.

**Methodology:** QinQin Yu, Joao A. Ascensao, Takashi Okada, Oskar Hallatschek.

**Project administration:** Oskar Hallatschek.

**Resources:** Oskar Hallatschek.

**Software:** QinQin Yu, Joao A. Ascensao, Takashi Okada.

**Supervision:** Erik Volz, Oskar Hallatschek.

**Validation:** QinQin Yu, Joao A. Ascensao, Takashi Okada, Olivia Boyd, Erik Volz, Oskar Hallatschek.

**Visualization:** QinQin Yu, Joao A. Ascensao, Takashi Okada, Olivia Boyd, Erik Volz, Oskar Hallatschek.

**Writing – original draft:** QinQin Yu, Joao A. Ascensao, Takashi Okada, Oskar Hallatschek.

**Writing – review & editing:** QinQin Yu, Joao A. Ascensao, Takashi Okada, Olivia Boyd, Erik Volz, Oskar Hallatschek.

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
