## [Decision Letter · Decision Letter 0]

14 Jun 2023

Dear Dr. Hallatschek,

Thank you very much for submitting your manuscript "Lineage frequency time series reveal elevated levels of genetic drift in SARS-CoV-2 transmission in England" for consideration at PLOS Pathogens. As with all papers reviewed by the journal, your manuscript was reviewed by members of the editorial board and by several independent reviewers. In light of the reviews (below this email), we would like to invite the resubmission of a significantly-revised version that takes into account the reviewers' comments.

The topic of this paper is a very interesting and relevant one. All three reviewers raise substantial concerns about different methodological issues, and reviewers 2 and 3 make many valuable suggestions on how to correct/clarify. While reviewer 1 is less positive, I believe these are issues that can be addressed by the authors. Specifically I would be happy to see how they consider purifying selection impacts the estimation of genetic drift.

We cannot make any decision about publication until we have seen the revised manuscript and your response to the reviewers' comments. Your revised manuscript is also likely to be sent to reviewers for further evaluation.

Sincerely,

Adi Stern

Academic Editor

PLOS Pathogens

Ronald Swanstrom

Section Editor

PLOS Pathogens

Kasturi Haldar

Editor-in-Chief

PLOS Pathogens

orcid.org/0000-0001-5065-158X

Michael Malim

Editor-in-Chief

PLOS Pathogens

orcid.org/0000-0002-7699-2064

The topic of this paper is a very interesting and relevant one. All three reviewers raise substantial concerns about different methodological issues, and reviewers 2 and 3 make many valuable suggestions on how to correct/clarify. While reviewer 1 is less positive, I believe these are issues that can be addressed by the authors. Specifically I would be happy to see how they consider purifying selection impacts the estimation of genetic drift.

Reviewer's Responses to Questions

**Part I - Summary**

Reviewer #1: see below

Reviewer #2: This manuscript uses UK COVID-19 survey data to estimate the "effective population size" of the SARS-CoV-2 epidemic in the UK, and finds that it is roughly 100x smaller than the number of cases. Concretely, the fluctuations in the observed frequencies of different lineages over time are far larger than expected. While some of the fluctuations are due to fluctuating biases in sampling, for the most part they compound over time, indicating that they are primarily due to fluctuations in the true underlying frequencies of the lineages. I think this paper is very exciting, both for its results and as an example of the new kinds of epidemiological analyses that can be done with large sequence surveys. Overall, I think that the scholarship is very solid, although I do I have some suggestions that the authors may want to consider.

I guess the issue with simultaneously inferring selection and drift is computational, in that it breaks the nice factorized form of the approximate likelihood. (As I mentioned, I think that the importance of this form should be pointed out earlier in the manuscript.) If this makes it impractical, then I think another option would be to follow the same s inference procedures as currently used, but discard lineages based on the magnitude of the point estimates of s or s * (delta t), where delta t is the time interval over which the lineage is observed, rather than a statistical significance test. Or one could even use a direct comparison of the inferred magnitude of allele change due to selection vs the expected allele frequency change based on the inferred N_e and keep only those lineages where the former was small compared to the latter.

Reviewer #3: The authors present an approach to infer genetic drift and measurement noise from time-series lineage frequency data using genomic sequence and surveilance data from COG-UK up to December 2021. Measurement noise is overdispersed and the strength of overdispersion varies over time. The authors show that these two types of noise leave distinct footprints in the data. The method is convincing, presented in a clear way, and validated on simulated data.

The inference for SARS-CoV-2 is performed in 4 distinct major lineages to limit the effect of selection. The authors map the effective population size as a function of time for each major lineage. They show that the inferred effective population size is orders of magnitude smaller than what one would expect from an SIR model using case data. The authors discuss two potential mechanism that might explain the high level of genetic drift: super-spreading and deme structure within the population.

The authors have used a careful method to infer an important evolutionary parameter of SARS-CoV-2. However, their results may be more strongly linked to the evolutionary context.

**Part II – Major Issues: Key Experiments Required for Acceptance**

Reviewer #1: Yu et al.

Reviewed for PLOS Pathogens

In this manuscript, the authors propose an approach to jointly estimate parameters related to genetic drift and to sampling variance using time-series data. They apply the approach to CoV-2 sequence collected in the UK in 2020-2021. They argue that the degree of observed genetic drift is higher than may be expected.

I see a number of population genetic oddities in this work. While the authors jump through many hoops in an attempt to justify the assumption of neutrality within lineages in order to simplify interpretations, these efforts are unsatisfactory. As degrees of genetic drift are being gleaned from changes in allele frequency, and given that the CoV-2 genome is essentially entirely functional, this neutrality assumption (even within strains) is unjustified. While the authors perform a simple test to examine deterministic allele frequency changes in an effort to identify 'possibly non-neutral' effects, this is problematic as the authors seem to be searching for the sort of deterministic increases in allele frequency associated with selective sweeps only. However, the vastly more common type of selection will be purifying in nature, and thus they will be unable to observe 'deterministic' changes in allele frequency amongst rare variants. As such, the vast majority of their sequence data will be impacted by purifying selection as well as the resulting background selection effects. They simply can not justify neglecting these major evolutionary contributors. This negative selection will of course be occurring in all 'strains' and 'lineages', and will to a first approximation give the appearance of a reduced effective population size. Given that the main conclusion of this manuscript is that genetic drift may be stronger than one would expect given "the number of positive individuals" observed (though the basis for this claim is still quite unclear to me), I see no reason to think that neglected purifying selection effects would not be the main contributor to the observation. The authors may want to study this recent AREES Review to consider these expectations (Charlesworth & Jensen, 2021, 'Effects of selection at linked sites on patterns of genetic variability'). As another of many possible examples in this regard, the authors also neglect gene flow, which is likely considerable in this human pathogen, and which would also serve to inflate estimates of genetic drift when ignored.

The authors propose other unsupported assumptions to further simplify, and justify this as "we consider the extreme situation where only one mechanism at a time is driving the dynamics". Again, there will simply be no empirical data in which only one evolutionary force is acting; at a bare minimum this CoV-2 data is being shaped by mutation, recombination, genuine genetic drift as driven by population size change and structure, purifying selection, background selection, 'superspreading', etc. The authors may want to consider the recent PLOS Pathogens Review on general evolutionary dynamics in CoV-2 (Terbot et al., 2023, 'Developing an appropriate evolutionary baseline model for the study of SARS-CoV-2 patient samples'). The authors may also want to consider this more generalized PLOS Biology Commentary, written by many of the leading thinkers in evolutionary inference, about the dangers of neglecting known-to-be-contributing evolutionary processes when performing such estimation (Johri et al., 2022, 'Recommendations for improving statistical inference in population genomics').

As a final point, it is entirely unclear why the authors are describing effective population size under a randomly mating Wright-Fisher model at all (and why test simulations in particular are being performed under this clearly inappropriate model), when they themselves are making a case for skewed offspring distributions via 'superspreading'. There are of course many developed frameworks under generalized Moran models and multiple merger coalescent theory that would be much more appropriate for a viral population of this sort.

In summary, the neglect of fundamental evolutionary processes and population genetic considerations makes this work unsuitable for publication.

Reviewer #2: 1. It would be nice to do more in the way of model checking, i.e., quantifying how well the fitted model matches the observed data. Some possible plots (maybe just as SI):

a. Scatter plot of $(\\sqrt{f_{t+1} - \\sqrt{f_t})$ vs $f(t)$.

b. Histogram of $(\\sqrt{f_{t+1} - \\sqrt{f_t}) / \\sqrt{\\tilde N_e(t)}$. This should be approximately normal, right?

c. Scaling of the mean squared change in the square root of frequency with the time elapsed. (Again, this could be normalized by the estimated $\\tilde N_e(t)$ to allow data from different times to be averaged together.) This should be linear, right? (And with a known slope.) Saturating behavior would correspond to measurement error, and super-linear behavior could be due to selection or correlations in local transmission network structure. Maybe this is too messy to say anything definitive, but even if that's the case, it's nice to show it. Another way to approach this might be with a scatter plot of $f_{t+2} - f_{t+1}$ vs $f_{t+1} - f_t$.

Just throwing ideas out there, but I do think it's important to have some figures somewhere that show what features of the data the model is capturing and what features might point to other dynamics beyond the model.

2. For removing spurious fluctuations that are actually due to selection, I don't understand the approach based on looking for lineages where one can reject a null hypothesis of neutrality. This would make sense if the goal of the paper were to identify lineages showing clear signs of selection for, e.g., follow-up on their characteristic mutations. But here what matters is how selection might bias estimates of drift, which is a different question. We know a priori that there are no lineages with a selective coefficient of exactly zero. I think it would make more sense to fit a model with each lineage having its own constant selective coefficient $s_i$ that gets inferred along with all the other parameters. As long as most of the lineages are observed for multiple time points, it seems like it should be possible.

Reviewer #3: The following points should be addressed in a revision:

1. A point of concern is how the selection between major lineages could impact the inference method and the results. This applies specifically to well known clade displacements Alpha-Delta (May/June 2021), and Delta-Omicron, which happens just after the data cut-off on Dec. 2021. In both cases, the ancestral clade has census population sizes dropping very quickly, and how this affects the inference remains to be tested in simulations. A point to be noted is that the crossover from Alpha to Delta, which corresponds to a bottleneck in census population sizes, shows up in a plausible way in the inferred N_e. However, no trace of the next crossover is to be seen in the reported N_e. The analysis should be extended at least to February 2022 to cover the full crossover from Delta to Omicron (BA.1).

2. The authors limit their study on the COG-UK case-data because they provide "consistently large number of sequences SARS-CoV-2 cases". There is a number of other regions/countries with good sequencing and surveillance of SARS-CoV-2 cases over time (e.g., California, Germany, France). Is the method easily extendable to countries other than England? The manuscript makes a point on scalability and applicability of the method, so validation beyond one showcase scenario seems appropriate.

3. Rare lineages that fall below the count/frequency threshold are randomly combined into "superlineages" that are above the threshold number. This may combine sequences that are far apart from each other on the tree instead of following topological structure. The procedure would be fine in a truly neutral evolution, but may affect a possible signal of selection. We would propose to combine rare lineages with their parent lineage instead. Also, the term "superlineage” is not very appropriate if the joined lineages are not closely related. Perhaps the term should be reconsidered anyway, because a reader could misunderstand it as suggesting higher fitness or super-spreading. Maybe ``coarse-grained lineages’’ is more appropriate?

4. The following is not a ``major issue'' but a suggestion on how to increase the impact of the paper.

At the end of the day, what do we learn from the decomposition of noise into (broadly interpreted) genetic drift and measurement noise? This question deserves more quantitative comments. The authors mention the origination probability for new mutant lineages, which is clearly

---

## [Decision Letter · Decision Letter 1]

15 Nov 2023

Dear Dr. Hallatschek,

Thank you very much for submitting your manuscript "Lineage frequency time series reveal elevated levels of genetic drift in SARS-CoV-2 transmission in England" for consideration at PLOS Pathogens. As with all papers reviewed by the journal, your manuscript was reviewed by members of the editorial board and by several independent reviewers. In light of the reviews (below this email), we would like to invite the resubmission of a significantly-revised version that takes into account the reviewers' comments.

The reviewers have seen the revised paper, and there are still major concerns. The main one is stressed by reviewer 3 - what does Ne mean at all when we are in a scenario where there may be strong selection (but - this can be argued as well). I think this is always a concern in the field of evolutionary biology, and most often we simply have less info than in the the COVID19 pandemic situation. Thus, any analyses or discussion that more directly address this point might be helpful - e.g., if the treatment reproduced more than one of the obvious bottlenecks in actual case numbers. Having said that, I appreciate this is a lot of extra work and the authors may want to consider another venue in such a case. Typically we would reject a paper that still raised concerns with the reviewers in a resubmission. While we have not taken that course here, it will be important to convince the reviewers that the significant questions raised have been addressed.

We cannot make any decision about publication until we have seen the revised manuscript and your response to the reviewers' comments. Your revised manuscript is also likely to be sent to reviewers for further evaluation.

Sincerely,

Adi Stern

Academic Editor

PLOS Pathogens

Ronald Swanstrom

Section Editor

PLOS Pathogens

Kasturi Haldar

Editor-in-Chief

PLOS Pathogens

orcid.org/0000-0001-5065-158X

Michael Malim

Editor-in-Chief

PLOS Pathogens

orcid.org/0000-0002-7699-2064

The reviewers have seen the revised paper, and there are still major concerns. The main one is stressed by reviewer 3 - what does Ne mean at all when we are in a scenario where there may be strong selection (but - this can be argued as well).

Having said that, I think this is always a concern in the field of evolutionary biology, and most often we simply have less info than in the the COVID19 pandemic situation. Thus, any analyses or discussion that more directly address this point might be helpful - e.g., if the treatment reproduced more than one of the obvious bottlenecks in actual case numbers. Having said that, I appreciate this is a lot of extra work and the authors may want to consider another venue in such a case.

Reviewer's Responses to Questions

**Part I - Summary**

Reviewer #2: The authors have done a ton of work responding to reviews and have addressed my concerns. Fig R9/S14 looks great!

Reviewer #3: (No Response)

**Part II – Major Issues: Key Experiments Required for Acceptance**

Reviewer #2: None

Reviewer #3: The paper presents an approach to separately infer the amount of genetic drift and measurement noise from genetic data. The main observation of the paper: the inferred effective population size (as defined in a Wright-Fisher model) is two orders of magnitude smaller than the number of positive cases. I appreciate the extensive work that has gone into the revision and the reply to reviewers’ comments. Nevertheless, the revised version leaves a number of questions open. 

 1. The authors claim that the method is highly scalable because it only relies on lineage frequency time series data. However, in other parts of the paper, the authors do explicitly rely on the phylogenetic tree. This constraint also appears to preclude the application to data from different countries. At the same time, the authors find it difficult to combine smaller clades with their parent lineages, which should not be a problem when a tree is available. Additionally, the authors mention that they cannot extend the analysis to later times than March 2022 because there is not enough data, which precludes the analysis of later bottlenecks. Indeed, the total number of available sequences for UK decreases from >150.000 per month to ~20.000 per month, which should still be sufficient for the kind of analysis performed. Together, it remains unclear to what extent the method is scalable, what amount of data is necessary, and whether the method only relies (and should rely) on lineage frequency time series.

2. The authors did not address the concern of how between-lineage selection affects the method of inference of effective population size. When a lineage is exponentially changing in size due to between-lineage selection, this could affect the inferred effective population size or its interpretation. Clearly, this point is related to the other referees’ points on within-lineage selection, e.g. purifying selection (which should be ubiquitous). 

In summary, the manuscript presents a clever idea of noise decomposition to obtain estimates of the effective population size in a highly complex evolutionary system far from equilibrium. At the same time, some methodological questions remain open. More importantly, the discussion with the reviewers has shown that the evolutionary scenario of this system is, at least, sufficiently far away from textbook neutrality that the very meaning of the effective population size, regarded as an intermediate summary statistics for the computation of measurable quantities, remains to be clarified. What falsifiable results could you or others compute from the pattern of effective population sizes found, in order to ultimately corroborate or refute the contentious assumption of partial neutrality made in the inference procedure? For example, should we expect more nonsynonymous (slightly deleterious) mutations originating in the bottleneck of June 2021 getting eventually fixed or not? How does the fixation probability of a mutation, in the scenario derived, depend on the sampled frequency of an actual observation? This type of questions is what I would hope any further revision of the paper could shed more light on.

**Part III – Minor Issues: Editorial and Data Presentation Modifications**

Reviewer #2: The one thing I would say is that I think Fig R10 is a very interesting result as well and would recommend also including it in the manuscript. But I can also see the argument that this manuscript is already turning into a beast, and maybe Fig R10 is sufficiently interesting that it's worth following up on in its own separate work rather than getting lost in this one.

Reviewer #3: (No Response)

PLOS authors have the option to publish the peer review history of their article (what does this mean?). If published, this will include your full peer review and any attached files.

Reviewer #2: **Yes: **Daniel B. Weissman

Reviewer #3: No
---

## [Decision Letter · Decision Letter 2]

3 Mar 2024

Dear Dr. Hallatschek,

We are pleased to inform you that your manuscript 'Lineage frequency time series reveal elevated levels of genetic drift in SARS-CoV-2 transmission in England' has been provisionally accepted for publication in PLOS Pathogens.

Best regards,

Adi Stern

Academic Editor

PLOS Pathogens

Ronald Swanstrom

Section Editor

PLOS Pathogens

Michael Malim

Editor-in-Chief

PLOS Pathogens

orcid.org/0000-0002-7699-2064

Both reviewers agree the authors have done an excellent job. Reviewer 2 has a minor suggestion - and specifies this is at the authors discretion.

Reviewer Comments (if any, and for reference):

Reviewer's Responses to Questions

**Part I - Summary**

Reviewer #2: This is a really impressive study.

Reviewer #3: Thanks to the authors for their patience with the reviewers’ comments. The current revision adds answers to the remaining questions and is, in my opinion, now ready for publication. The data limitations are discussed, and the method is established as a proof of principle for the inference of an effective population size in highly diverse viral systems. I am looking forward to applications of this method to other systems with more long-term datasets in future work. For now, I would encourage you to put the prediction of a consequence of time-dependent N_e, namely a specific time-dependent pattern of establishment rates, slightly more upfront (and perhaps in the abstract). The published paper will be read by people who believe that this system is a travelling fitness wave and N_e is an obsolete concept. The clearest answer you can give to them, in terms of falsifiable statements, will be appreciated by that readership. (But this is all at the discretion of the authors.)

**Part II – Major Issues: Key Experiments Required for Acceptance**

Reviewer #2: (No Response)

Reviewer #3: (No Response)

**Part III – Minor Issues: Editorial and Data Presentation Modifications**

Reviewer #2: (No Response)

Reviewer #3: (No Response)

PLOS authors have the option to publish the peer review history of their article (what does this mean?). If published, this will include your full peer review and any attached files.

Reviewer #2: No

Reviewer #3: No

---

## [Editor Report · Acceptance letter]

2 Apr 2024

Dear Dr. Hallatschek,

We are delighted to inform you that your manuscript, "Lineage frequency time series reveal elevated levels of genetic drift in SARS-CoV-2 transmission in England," has been formally accepted for publication in PLOS Pathogens.

Best regards,

Michael Malim

Editor-in-Chief

PLOS Pathogens

orcid.org/0000-0002-7699-2064